# Impacts of solar intermittency on future photovoltaic reliability

Jun Yin 🄳 [1], Annalisa Molini 🄳 [2,3] & Amilcare Porporato 🄳 [4,5 ✉]

As photovoltaic power is expanding rapidly worldwide, it is imperative to assess its promise under future climate scenarios. While a great deal of research has been devoted to trends in mean solar radiation, less attention has been paid to its intermittent character, a key challenge when compounded with uncertainties related to climate variability. Using both satellite data and climate model outputs, we characterize solar radiation intermittency to assess future photovoltaic reliability. We find that the relation between the future power supply and long-term mean solar radiation trends is spatially heterogeneous, showing power reliability is more sensitive to the fluctuations of mean solar radiation in hot arid regions. Our results highlight how reliability analysis must account simultaneously for the mean and intermittency of solar inputs when assessing the impacts of climate change on photovoltaics.

[1] School of Hydrology and Water Resources, Nanjing University of Information Science and Technology, Nanjing 210044, China. [2] Masdar Institute, Khalifa University of Science and Technology, PO Box 54224 , Abu Dhabi, UAE. [3] Department of Civil Infrastructure and Environmental Engineering, Khalifa University of Science and Technology, PO Box 127788 , Abu Dhabi, UAE. [4] Department of Civil and Environmental Engineering, Princeton University, Princeton, NJ 08544, USA. [5] Princeton Environmental Institute, Princeton University, Princeton, NJ 08544, USA. ✉email: aporpora@princeton.edu

Increasing the use of solar energy is widely regarded as one of the most effective approaches to reduce $CO_2$ emissions, yet the short-term intermittent nature imposes definite limitations to its reliability. While this problem may be partially solved by power storage, geographic dispersion, load control, and radiation forecasting[1–3], it still has significant impacts on the grid integration of solar energy. For instance, photovoltaic power plants in Northwestern China (capacity of 43.87 GW in 2019, 1/3 of China's total) were punished for providing intermittent energy to the Northwest Grid with fines of $28 million US dollars in 2017, $42 million in 2018, and $28 million for the first half of the year 2019, whereas coal-fired and hydropower plants were rewarded for their constant and even dispatchable sources of electricity[4–6]. Similarly, the example of Kauai island, Hawaii, a world pioneer in using renewable energy[7], currently relies on diesel generators on overcast days[8,9]. While the solar radiation varies across a range of timescales, here we focus on the daily level, which accounts for a significant portion of the penalty in the case of the Northwestern China[4,5] and is closely related to the power reliability in Kauai, Hawaii[9].

The daily radiation is expected to change in future climates due to altered cloud and aerosol patterns[10–14], presenting additional challenges for the long-term planning and management of solar energy. Previous studies have focused mostly on the relative change of long-term mean radiation input[15–19]. While mean metrics are essential, the portion of time with energy supply lower than the demand, termed loss-of-load probability (LOLP)[20], which is related to the reliability and the market values of power output, cannot be captured by mean values alone. Power reliability is vital for grid planning and management. For example, the solar plant from Tesla is expected to provide 52 MWh of electricity every evening to the power grid in Kauai, Hawaii[7]. Tesla's design of 13 MW solar array and 52 MWh effective battery storage result in an LOLP of 0.12, possibly maximizing the net profit while still satisfying the reliability requirement[9]. In a grid-connected system, LOLP is directly associated with the operating cost of the peaking plants (e.g., diesel generators in Kauai, Hawaii[8,9], hydropower stations in Northwest of China[21], gas turbines in the Great Plains, United States[22]) and thus linked to the market values of the solar energy.

To investigate the impacts of future climates on LOLP, we combine here satellite-derived data and climate model outputs. In particular, we focus on the impact of incident solar irradiance, one of the dominant factors controlling solar power generation[15,17,18]. We show the nonlinear behaviors of LOLP in response to climate change, pointing towards a tradeoff between the potential power outputs and the power reliability.

## Results

**Characterizing solar energy intermittency**. We begin our investigation with an analysis of the clearness index, $K$, defined as the ratio between the near-surface global horizontal irradiance (GHI, including direct and diffuse irradiance) and the corresponding extraterrestrial horizontal irradiance (see "Methods" section). This index accounts for the scattering, absorption, and reflection of solar radiation from all optically active constituents in the atmosphere, such as clouds and aerosols, and is often used in solar energy industry[23–26]. For example, we consider Southeastern Romania's case, where climate change has shown strong regional impacts[27] and the case of Dubai, UAE, which is pursuing an ambitious plan to foster solar energy development in the region[28]. Romania and UAE, located in the continental and desert climatic zones, also have two contrasting cloud seasonality (see Supplementary Fig. 1) and drastically different solar energy production potentials. We use satellite data from Clouds and the

Earth's Radiant Energy System (CERES), which are based on column-model estimates and have been already used for solar power assessment[29,30]. Such multi-decadal records allow us to characterize the empirical distributions of daily $K$. As can be seen in Fig. 1, the $K$ distributions for larger mean values (denoted as $\mu$ and also referred to as the mean clearness index) tend to have longer left tails, which are associated with the weaker solar radiation and lower power generation.

From the $K$ distribution, the LOLP of a solar power plant operating at daily basis (e.g., the Tesla's power plant at Kauai, Hawaii) can be estimated as the fraction of days with solar radiation lower than the demand value,

$$\text{LOLP} = \int_0^{K_D} f(K)dK, \qquad (1)$$

where $f(K)$ is the probability density function (pdf) of $K$, and $K_D$ is the value of $K$ that is just sufficient to meet the energy demand (see "Methods" section). LOLP is, therefore, the cumulative density function (CDF) of $K$ at $K_D$. This metric has long been used for designing a stand-alone (off-grid) photovoltaic power system[31–33] and is also a critical reference for evaluating a grid-connected system[20]. The constant demand $K_D$ in (1) is similar in spirit to the regulation from Northwest Grid of China, which was originally issued for coal plants considering their relatively constant power output but was recently extended to solar and wind power plants. A thorough characterization of the global solar power intermittency and its response to climate change using the LOLP is a fundamental starting point to assess the future reliability of photovoltaic.

Climate-change impacts on power reliability can be assessed by considering the change of LOLP during the lifespan of typical photovoltaic modules. Going back to the case of the Southern Romania, a solar plant designed under historical climate records of 2001–2009 is assumed to have a design LOLP, $\text{LOLP}_D$, of 0.3. Over the following nine years (2010–2018), the mean of $K$ increases in both January ($\Delta\mu = 0.015$) and July ($\Delta\mu = 0.03$), which may be associated with the change of climate seasonality[34]. The corresponding values of LOLP drop from the design value of 0.3 to 0.27 in winter ($\Delta\text{LOLP} = -0.03$) and to 0.21 in summer ($\Delta\text{LOLP} = -0.09$), respectively (see the hatched and shaded areas in Fig. 1a, b). For the case in Dubai, aerosol optical depth trends[35] may account for the increase of $\mu$ in winter, leading to a decrease of LOLP (Fig. 1c), while the monthly mean clearness index remains relatively constant in summer (Fig. 1d). The comparisons between these two periods (2001–2009 and 2010–2018) objectively quantify not only the increase in mean surface solar radiation, but also the increase in its reliability.

With this methodology, we now move to the future climate scenarios and use climate model outputs (see "Methods" section) to calculate the changes of $\mu$ and LOLP between 2006–2015 and 2041–2050, consistently with the typical lifespan of photovoltaic modules. As shown in Fig. 2a, b and in agreement with previous studies[15], the change of solar radiation is evident in some regions and show marked seasonal variations. The solar radiation in Europe is projected to decrease in January and increase in July, which may be associated with the projected changes in rainfall seasonality and the corresponding cloud variations[34]. The decrease in solar radiation in the Middle East may be associated with large-scale circulation[36], cloudiness trends[37], or the positive trends of aerosol optical depth as documented over large parts of the Middle East for the period 2001–2012[35].

This redistribution of the Earth's energy and shifts in climate seasonality[38] have direct impacts on solar power reliability as quantified by the corresponding variations of LOLP (see Fig. 2c, d). Although it is apparent that increasing solar radiation ($\Delta\mu > 0$)

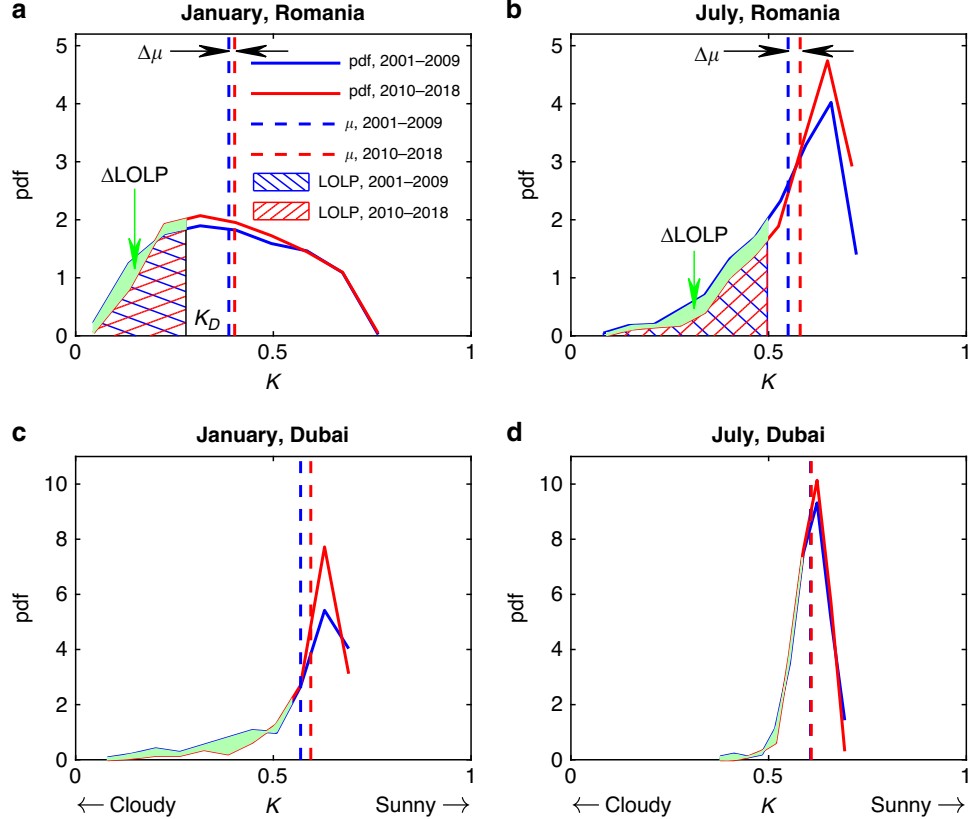

**Fig. 1 Examples of climate impacts on solar radiation and photovoltaic power reliability.** The distribution of clearness index ($K$) derived from the CERES data in (**a**, **c**) January and (**b**, **d**) July during 2001–2009 (blue lines) and during 2010–2018 (red lines) in (**a**, **b**) Southern Romania and (**c**, **d**) Dubai. The hatched areas indicate the probability when power generation does not meet the demand, the loss-of-load probability (LOLP). The averages of clearness index are marked by the vertical dash lines and the values are reported in Supplementary Table 1. Source data are provided as a Source Data file.

often leads to more reliable power output ($\Delta\text{LOLP} < 0$), this relationship is clearly nonlinear. For example, the slight decrease of solar radiation in the Middle East and Northern Africa results in a significant increase of LOLP; an increase of solar radiation in the west of Amazon rainfall forest in July leads to a sharp decrease of LOLP; strong variations in both radiation and power reliability are shown in the Northern United States in January. In what follows, we will investigate this nonlinear relationship to quantitatively link the previous reports on mean solar radiation to one of our major concerns on power reliability.

**Theoretical framework for power reliability.** The case studies in Fig. 1 and geographical patterns in Fig. 2 suggest that LOLP may be linked to the distribution of $K$, which in the solar industry is often associated with the mean clearness index, $\mu$[39,40]. To systematically and theoretically assess this linkage, we consider in detail satellite data as well as climate model outputs under the historical climate conditions. We obtained the statistics of $K$ from all regions over the world with $\mu$ ranging from 0.3 to 0.7 with a binning interval of 0.05 (see dark color curves in Fig. 3a and Supplementary Fig. 7). As can be seen, $f(K)$ tends to be positively skewed in regions with smaller $\mu$ and negatively skewed in regions with larger $\mu$ (see Fig. 3a). Since the diffuse radiation has the largest variations for moderate $K$[39], which includes direct and diffuse radiation, it is logical to expect $\sigma$ first increases and then decreases with rising $\mu$ as presented in Fig. 3b. Overall, such empirical distributions even under changing climate conditions turn out to be well described by beta distributions (see "Methods" section).

One may wonder whether these characteristics can vary in response to changing climates. To address this point, we checked

the statistics of $K$ at different periods (see the light-color curves in Fig. 3a, Supplementary Figs. 7 and 8). The results show that the distributions of $K$ appear identical and the $\mu \sim \sigma$ relationships almost remain unchanged. These behaviors essentially describe how the intermittency of solar radiation (i.e., $\sigma$) will adjust after the change of mean solar radiation (i.e., $\mu$), providing valuable information for solar power planning and management.

The invariant characteristics of $K$ allow us to link $\Delta\mu/\mu$ to $\Delta\text{LOLP}$ between different periods and thus, in turn, to obtain power-reliability information from previous reports on long-term mean solar radiation. Operationally, this can be accomplished by Taylor expanding Eq. (1) to first order as

$$\Delta\text{LOLP} \approx \underbrace{\mu \frac{\partial\text{LOLP}}{\partial\mu}}_{L_s}\left(\frac{\Delta\mu}{\mu} \times 100\%\right), \quad (2)$$

where $L_s$ is the sensitivity of LOLP to $\mu$ and can be derived analytically for the beta distribution of $K$ (see "Methods" section), and the change of $\mu$ in percentage format is usually consistent with other reports. In Eq. (2), the first term evaluates the climate impacts in terms of LOLP, whereas the term in the bracket assesses the future solar radiation in the conventional approach[15–19]. The relation between the two, $\Delta\text{LOLP}$ and $\Delta\mu/\mu$, is clearly associated with the sensitivity parameter $L_s$, a nonlinear function of $\mu$ and $K_D$ (or design LOLP, see Eq. (11) in "Methods" section). Particularly interesting is the fact that the absolute values of $L_s$ are larger in sunny regions/seasons with larger $\mu$ (see Fig. 4a). This may be accounted for by the fact that the small perturbation of $\mu$ in sunny regions tends to have larger change in the variability of solar radiation (i.e., large absolute values of

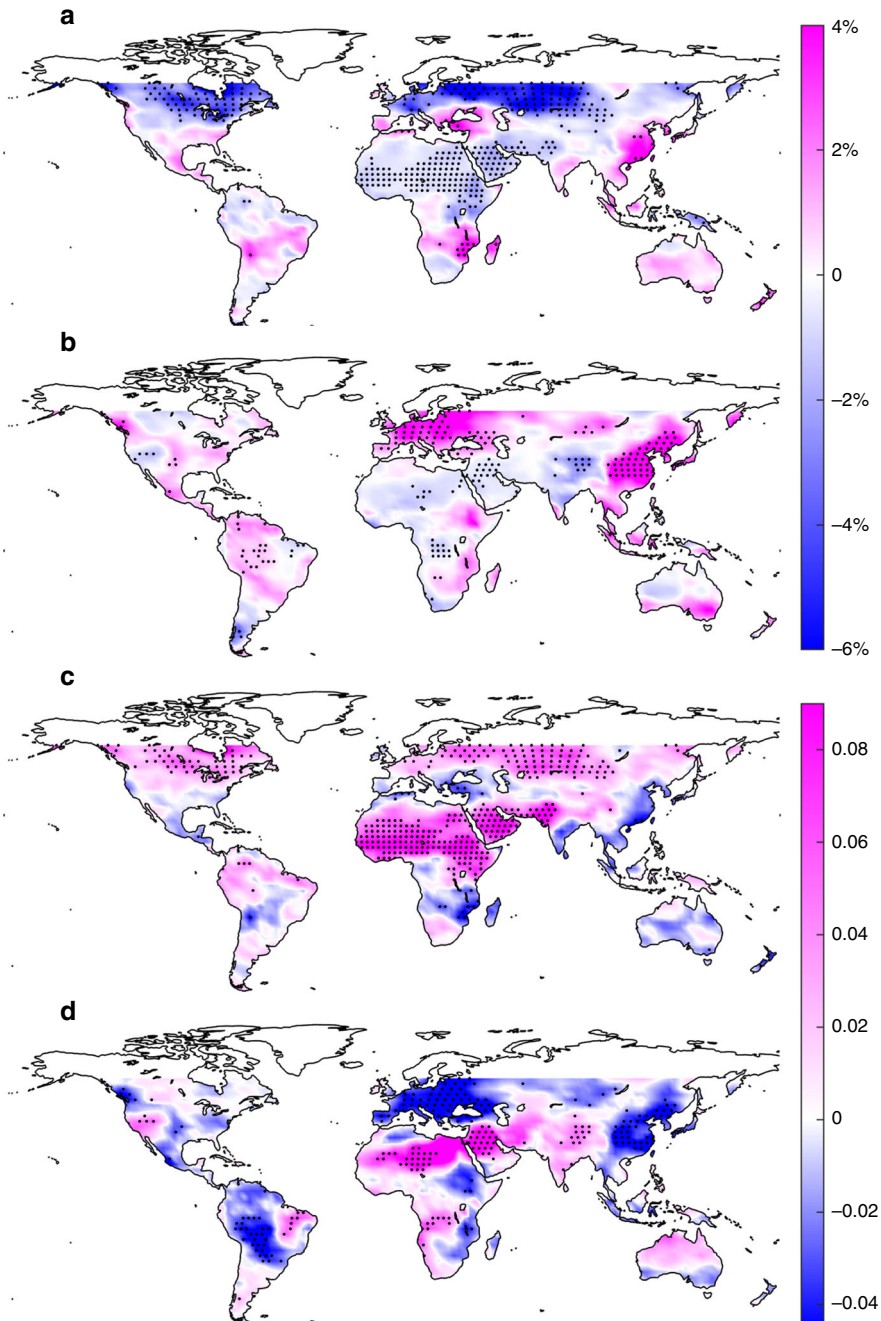

**Fig. 2 Variations of solar radiation and solar power reliability predicted from climate models.** The color at each grid point represents the ensemble means of (**a**, **b**) the relative change of mean clearness index ($\Delta\mu/\mu$) and (**c**, **d**) the change of loss-of-load probability ($\Delta$LOLP) between 2006–2015 and 2041–2050 in the month of (**a**, **c**) January and (**b**, **d**) July from 11 climate model outputs. The LOLP during 2006–2015 (i.e., design LOLP) is set as 0.3; maps with other design LOLP show similar patterns (see Supplementary Figs. 2 and 3). The dots show the ensemble mean of the corresponding variables are statistically different than zero, suggesting consistent variations of solar radiation or reliability from most climate models (*t*-test, 5% significance level; statistics of the sign of the changes are given in Supplementary Figs. 4-6). Source data are provided as a Source Data file.

$d\sigma/d\mu$, see right side of Fig. 3c), which is obviously associated with the intermittency of solar energy. Since these are also the regions of the world where the largest solar plants are expected to be deployed in the future, this fact should be considered with great attention in reliability analysis.

Climate model outputs corroborate the previous analytical results. Figure 4d shows $\Delta\mu/\mu$ and $\Delta$LOLP between 2006–2015 and 2041–2050 for given values of $\mu$ and design LOLP. The slopes of these two quantities are reported in Fig. 4b, showing similar patterns as their analytical counterparts (Fig. 4a).

With the obtained nonlinear function of $L_s$, one can readily infer the power reliability. To facilitate this, we mapped the analytical solution of $L_s$ in Fig. 4a to each location over the world with monthly mean clearness index from CERES data (see Fig. 5 and Supplementary Figs. 9 and 10). These maps could serve as lookup tables to assess power reliability in future climates. For example, Fig. 5 shows that $L_s$ is approximately −0.8 in January and −1.6 in July in Southern Romania for a design LOLP of 0.3. The mean solar radiation in this region is projected to vary around −15~0% in winter and around −5~5% in summer toward

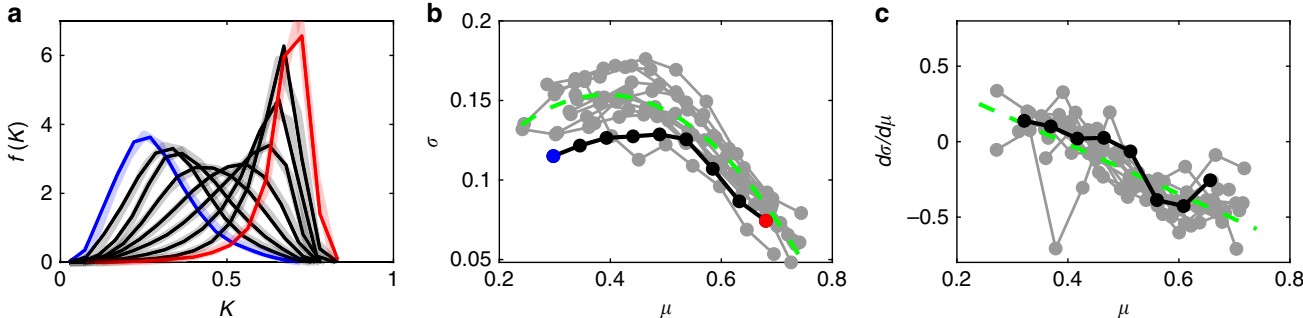

**Fig. 3 Statistics of the clearness index. a** Probability density functions (pdf) of daily clearness index ($K$) in different regions over the world (binning width of 0.05) from the satellite data in January during 2001–2009 (dark color) and during 2010–2018 (light color). **b** Relationship between mean ($\mu$) and standard deviation ($\sigma$) of daily $K$. The black/blue/red dots correspond to the lines in the **a**; the grey dots are from 11 climate model outputs during 2006–2015; the dash green curve shows the best quadratic fit. **c** $d\sigma/d\mu$ calculated as the derivative of the corresponding $\sigma \sim \mu$ relationship in (**b**). Source data are provided as a Source Data file.

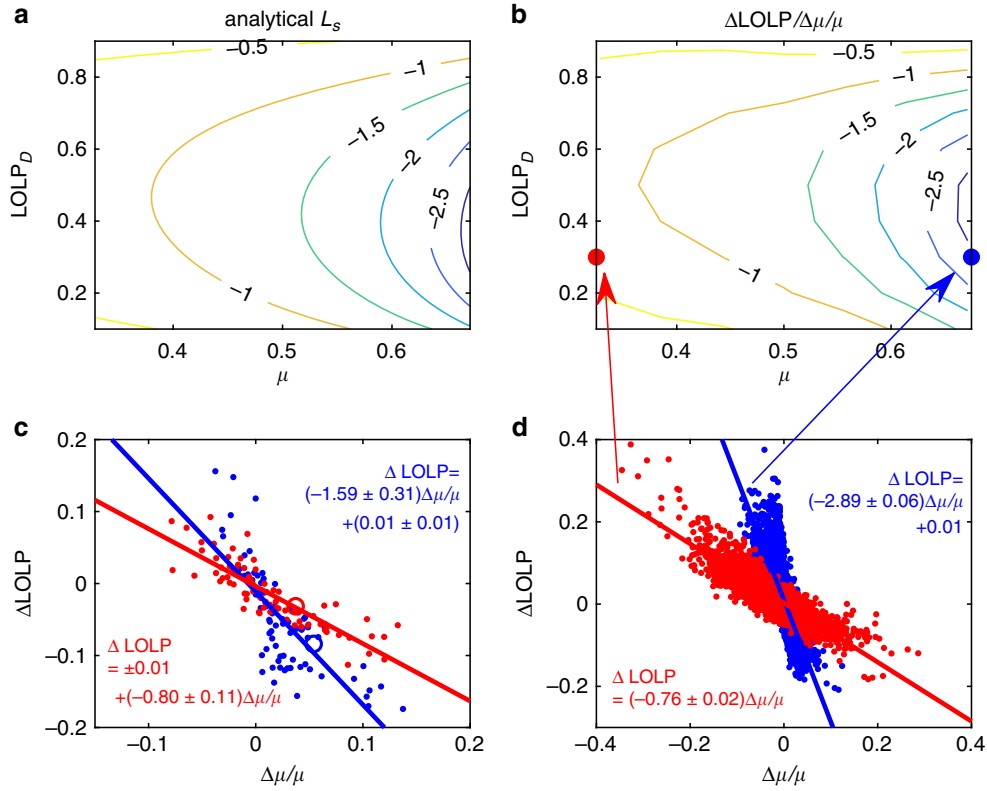

**Fig. 4 Sensitivity of loss-of-load probability ($L_s$).** Contour plots of $L_s$ is calculated (**a**) analytically from Eq. (2) and (**b**) numerically from climate model outputs. The red and blue dots in (**b**) are corresponding to the examples in (**d**), which compares the change of loss-of-load probability (LOLP) and the change of mean clearness index ($\mu$) from 2006–2015 to 2041–2050 in January with design LOLP of 0.3 in regions where $0.3 < \mu < 0.35$ (red dots) and $0.65 < \mu < 0.7$ (blue dots) as projected by climate models. The red and blue lines are the corresponding best fit lines and their slopes (i.e., $\Delta$LOLP / ($\Delta\mu/\mu$)) numerically represent $L_s$. (**c**) As in (**d**) but only for Bulgaria, Cyprus, Greece, Hungary, and Romania (i.e., region 7 defined in ref. [18]) in January (red dots) and July (blue dots). The red and blue circles correspond to the example of Southern Romania in Fig. 1. Source data are provided as a Source Data file.

the end of the century[18]. Multiplying these variations by $L_s$, one can find the impacts of these variations on LOLP (i.e., 0~12% in winter and −8~8% in summer). While the winter season has larger variations in solar radiation, it also has a small absolute value of $L_s$ so that the impacts on future power reliability in winter are reduced. This analysis is corroborated by the results from climate-model outputs as shown in Fig. 4c, which suggests larger spread of $\Delta$LOLP but slightly smaller change of $\Delta\mu/\mu$ in summer in the surrounding of Romania.

## Discussion

The heterogeneous distribution of LOLP sensitivity in Fig. 5 essentially stems from the nonlinear relationship between $\mu$ and $\sigma$, which remains relatively constant under changing climates. Lower absolute values of $L_s$ with smaller clearness index suggest that the solar power in humid subtropics may have lower potential for large variability in future climates. This is consistent with the observed $\sigma \sim \mu$ relationship in Fig. 3b, where these slopes are flatter for smaller $\mu$. Meanwhile, the humid subtropics are

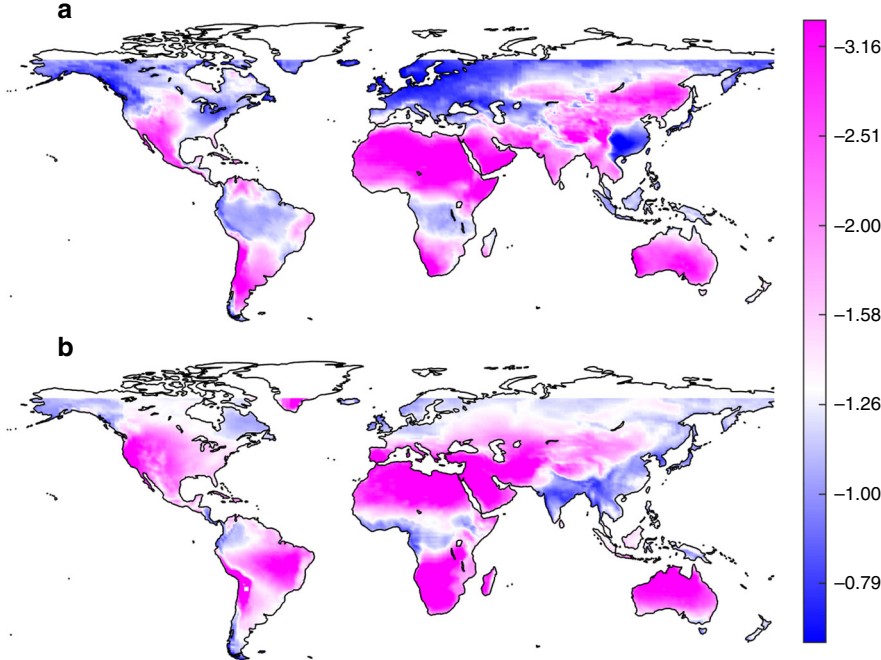

**Fig. 5 Global maps of LOLP sensitivity ($L_s$).** This sensitivity in (**a**) January and (**b**) July is obtained from analytical solutions with design LOLP of 0.3 and solar radiation climatology from CERES. Source data are provided as a Source Data file.

predicted to have relatively more solar radiation in the future climate scenarios[15,17]. The multiplication of small negative $L_s$ with positive $\Delta\mu$ yields small but negative $\Delta$LOLP, suggesting slightly higher power reliability. On the other hand, the arid hot regions are predicted to have less solar radiation but could yield much lower power reliability due to the strong LOLP sensitivity.

Our proposed framework may be further extended to diagnose the impacts of power storage, which is regarded as one of the most important solutions to the intermittency problems. Power storage smooths the power output to provide reliable energy. In our analysis, this effect may be considered by reducing the daily variability of future solar radiation and evaluating its impacts on LOLP (see "Methods" section). As expected, solar radiation with reduced variability has smaller LOLP, showing that increased storage can be used to mitigate the intermittency's impacts in most parts of the world. However, this may not be sufficient in a few regions such as the Middle East (see Supplementary Fig. 11). When mean solar radiation is significantly reduced as predicted by climate models, it may require increasing both the power storage capacity and solar module size.

To investigate more detailed grid operation and conduct cost-benefit analysis of various mitigation strategies, the proposed framework may be extended by statistically downscaling the daily solar radiation to the hourly timescale[39,41] and involving multiple power sectors for power generation, storage, transmission, distribution, marketing, and technology development[1,2]. Our framework could also be used to analyze the temperature impacts on power reliability (see "Methods" section), although it is argued that the temperature impacts on photovoltaic power generation appear much weaker than the solar radiation impacts over the lifespan of photovoltaic modules[36].

In summary, our results have shown how the impacts of this radiation change on power reliability could be significant due to the large absolute values of LOLP sensitivity, which had not emphasized previously. The sensitivity analysis points towards a tradeoff between the mean solar radiation that quantifies the total potential solar power and the power reliability, which being related to intermittency remains a major concern in the absence

of large power storage options. This contrasting behavior between solar power availability and reliability requires special attention in assessments of future solar energy scenarios.

## Methods
**Clearness index ($K$).** The daily clearness index, $K$, is defined as

$$K = \frac{\int_0^T \mathrm{GHI}(t)dt}{\int_0^T \mathrm{EHI}(t)dt}, \tag{3}$$

where $T$ is the length of 1 day, GHI is the near-surface global horizontal irradiance, which is the sum of the direct and diffuse irradiance, and EHI the extraterrestrial horizontal irradiance. Daily GHI are obtained from CERES SYN1deg during 2001–2018 and from 11 climate model outputs (ACCESS1.3, BCC-CSM1.1 m, CanESM2, CCSM4, CMCC-CMS, CSIRO-Mk3.6.0, EC-EARTH, GFDL-CM3, INM-CM4, IPSL-CM5A, and MPI-ESM) in "rcp45" experiment during 2006–2015 and 2041–2050. All these data have been used to obtain the empirical distributions of $K$ for calculating the loss-of-load probability as explained next.

**Loss-of-load probability (LOLP).** The photovoltaic power output is related to the incident solar radiation and other factors controlling the solar cell efficiency[15]. Each month, the Sun's declination angle has small variations; the daily incident solar radiation on a fixed or tracking array can be approximated as a monotonic function of daily clearness index[42]. Factors such as soiling and tree shading on solar modules could have notable impacts on power generation but can be controlled by regular maintenance. The solar cell efficiency factors such as air temperature and wind speed usually have only secondary impacts and are discussed in the following "Methods" section. Regarding climate change impacts, the incident solar radiation has been identified as the dominant factor for photovoltaic power generation. For this reason, we model the power output as a monotonic function of the clearness index, say $p = g(K)$. This function can be used to estimate the LOLP. Similarly to the off-grid version of a photovoltaic software[43,44], LOLP can be defined as the fraction of days when daily energy supply ($p$) is lower than the daily demand ($p_D$). We obtain LOLP as the derived distribution of $K$,

$$\mathrm{LOLP} = F(p_D = g(K_D)) = F(K_D) = \int_0^{K_D} f(K)dK, \tag{4}$$

where $K_D$ is the specific value of $K$ that is just enough to generate the demanding energy $p_D$, $f(\cdot)$ and $F(\cdot)$ are the probability and cumulative density function of $K$. These functions are estimated from multi-year historical climate records, and thus the corresponding LOLP already captures the interannual variability of daily power generation. Such estimates are referred to as design LOLP, $\mathrm{LOLP}_D$. For the lifespan

of typical photovoltaic modules (20–30 years), one can then quantify the climate impacts on power reliability as the change of LOLP from its design value.

**LOLP sensitivity ($L_s$).** The distributions of $K$ enters the LOLP expression in Eq. (4). As presented in Fig. 3a, the distribution of $K$ tends to be positively skewed for smaller mean value of $K$ (denoted as $\mu$) and negatively skewed for larger $\mu$. These behaviors may be described as beta distributions naturally bounded between 0 and 1. This is confirmed by the results of the Kolmogorov-Smirnov goodness-of-fit tests over most regions in the world in different climate zones (see Supplementary Fig. 12 and Supplementary Table 2)

$$f_b(K; \beta_1, \beta_2) = \frac{\Gamma(\beta_1 + \beta_2)}{\Gamma(\beta_1)\Gamma(\beta_2)} K^{\beta_1 - 1}(1 - K)^{\beta_2 - 1}, \quad (5)$$

where $\beta_1$ and $\beta_2$ are the shape parameters. Note that this beta distribution is a parsimonious choice which we prefer to other unbounded distributions (e.g., Weibull and extreme value distributions) used in the literature[45,46]. We stress however that our framework is not limited to the use of beta distributions but can easily adopt other distributions if they appear more suitable in some regions (e.g., Australia and Western Sahara). These shape parameters can be expressed by the mean ($\mu$) and standard deviation ($\sigma$) of the distribution[47],

$$\beta_1 = \frac{\mu(\mu - \mu^2 - \sigma^2)}{\sigma^2}, \quad (6)$$

and

$$\beta_2 = \frac{(1 - \mu)(\mu - \mu^2 - \sigma^2)}{\sigma^2}. \quad (7)$$

As described in Fig. 3b, the standard deviation may be modeled as a function of mean (e.g., $\sigma = -0.83\mu^2 + 0.65\mu + 0.03$, the best quadratic fit) so that the distribution of $K$ can be written as

$$f_b(K; \beta_1, \beta_2) = f_b(K; \beta_1(\mu, \sigma(\mu)), \beta_2(\mu, \sigma(\mu))). \quad (8)$$

Substituting (8) into (4) and performing a Taylor expansion to first order yields

$$\Delta\text{LOLP} \approx \underbrace{\mu \frac{\partial\text{LOLP}}{\partial\mu}}_{L_s}\left(\frac{\Delta\mu}{\mu} \times 100\%\right), \quad (9)$$

where

$$L_s = \mu\left(\frac{\partial\sigma}{\partial\mu}\frac{\partial\beta_1}{\partial\sigma} + \frac{\partial\beta_1}{\partial\mu}\right)\frac{\partial F}{\partial\beta_1}\bigg|_{K=K_D} + \mu\left(\frac{\partial\sigma}{\partial\mu}\frac{\partial\beta_2}{\partial\sigma} + \frac{\partial\beta_2}{\partial\mu}\right)\frac{\partial F}{\partial\beta_2}\bigg|_{K=K_D}, \quad (10)$$

where $F_b(\cdot)$ is the cumulative beta distribution and $K_D$ is equivalent to design LOLP,

$$\text{LOLP}_D = F_b(K_D). \quad (11)$$

The corresponding analytical solutions of $L_s$ (Fig. 4a) are very similar to its counterpart calculated numerically as $\Delta\text{LOLP}/(\Delta\mu/\mu)$ (Fig. 4b). The approximation of Taylor expansion to the first order is justified by the fact that $L_s$ is relatively constant for a small perturbation of $\mu$ (see Fig. 4a). Clearly, one can insert other distributions suitable in some specific regions into Eqs. (4) and (9) to obtain the corresponding analytical expression for the sensitivity of power reliability.

**Impacts of temperature change on power reliability.** Temperature influences the energy conversion efficiency and can have significant impacts on power generation in hot climates[48]. It is estimated that photovoltaic power output reduces by 0.45% for each degree increase in temperature[49,50]. Therefore, we may treat the temperature rising as equivalent to the increase of power requirement in our original framework and redefine the parameter $K_D$ as

$$K_D = [1 + \gamma_T(T - T_r)]K_{D,r}, \quad (12)$$

where the temperature factor, $\gamma_T$, is about 0.0045/K, $T_r$ is the reference temperature, and $K_{D,r}$ is the specific value of $K$ that is just enough to generate the demanding energy at the reference temperature. With this change, the corresponding LOLP becomes,

$$\text{LOLP} = \int_0^{[1 + \gamma_T(T - T_r)]K_{D,r}} f(K)dK. \quad (13)$$

The change of LOLP from current to future climate conditions can be expressed as

$$\Delta\text{LOLP} \approx \underbrace{\mu\frac{\partial\text{LOLP}}{\partial\mu}}_{L_s}\left(\frac{\Delta\mu}{\mu} \times 100\%\right) + \underbrace{T\frac{\partial\text{LOLP}}{\partial T}}_{L_T}\left(\frac{\Delta T}{T} \times 100\%\right), \quad (14)$$

where

$$L_T = TK_{D,r}\gamma_T f(K_D). \quad (15)$$

This expression suggests that the change of LOLP has two parts. This first part is in Eq. (9) and the second part can be obtained analytically by substituting Eq. (5)

into Eq. (15). The sensitivity for temperature, $L_T$, is always positive (see Eq. (15)), meaning that rising temperature increases the LOLP.

**Impact of power storage on power reliability.** Power storage at multiday timescale, if feasible, would obviously help improve power reliability. To explore this issue within the scope of the present analysis, as a proof of concept, we simply smoothed the daily clearness index to roughly estimate the impacts of power storage on power reliability

$$K_b = \mu + b(K - \mu), \quad (16)$$

where the clearness index $K$ is smoothed into $K_b$. The corresponding standard deviation becomes

$$\sigma_{K_b} = b\sigma, \quad (17)$$

where the coefficient $b$ controls the reduction of the variability. This coefficient $b$ is set as 0.75 and 0.5 for two future scenarios corresponding to the 25 and 50% variability mitigation.

We applied Eq. (17) to recalculate the clearness index from 11 climate model outputs during 2041–2050, which were then used to numerically calculate the LOLP. We showed the change of LOLP with no variability mitigation, 25% mitigation, and 50% mitigation in Supplementary Fig. 11. Reducing the variability leads to a decrease of LOLP and thus more reliable power output as expected. This is generally sufficient for addressing some of the challenges of intermittent solar power and the uncertainties related to climate change. In some regions, however, climate models also predict decreasing trends of mean solar radiation, which may not be compensated only by the power storage. This is the case of the Middle East, where solar power is projected to be significantly reduced, so that LOLP increases even with variability mitigation measures (see Supplementary Fig. 11).

**Data accuracy.** To provide information regarding the data accuracy, we compared these satellite data and climate model outputs with the data from National Solar Radiation Database (NSRDB). The latter are produced by ground observations, satellite data, and meteorological models and are arguably one of the most reliable datasets for assessing the long-term spatial and temporal variability of the solar resource[51]. It should be noted that validating the global solar irradiance and surface energy balance is one of the biggest challenges in the climate science community[52,53].

Two typical outputs with different assimilation models, METSTAT and SUNY, are achieved in NSRDB [https://rredc.nrel.gov/solar/old_data/nsrdb/] and both are recommended by NREL. We compared SUNY and METSTAT during 2001–2010 when both products are available (see Supplementary Fig. 13a, b). Of 1415 sites over the United States (sites with missing data are excluded), the root mean square errors (RMSE) between these two outputs are around 0.05, which may be considered as the systematic biases from NSRDB. When further compared these measurements with satellite data (CERES SYN) in the same locations during the same period (see Supplementary Fig. 13c–f), one finds similar ranges of RMSE, suggesting that the satellite products are as accurate as these reliable data.

We then compared the long-term clearness index from the satellite data and the climate model outputs during 2006–2015 averaged at 280 km equal-area grids over the world (see Supplementary Figs. 14, 15, and Supplementary Table 3). The RMSE for some climate models (e.g., CCSM, GFDL) are similar to these SUNY-METSTAT differences from NSRDB, while for others the RMSE is at least of the same order of magnitude.

Besides these data comparison, it is also important to note that aerosol is a key climate component and future aerosol emissions are usually described as different scenarios such as Representative Concentration Pathways (RCPs)[54]. Our results are from RCP45, which includes the projected decline in aerosols during the 21th century because of the emission controls[55]. While the future aerosol emissions are prescribed, not all models include their indirect effects related to the aerosol-cloud interaction (see Supplementary Table 4), which could have an impact on cloud formation and the prediction of solar radiation[56]. However, these indirect effects do not seem to have strong impacts on the relationship between the mean and standard deviation of the radiation (see Supplementary Fig. 8), a key feature in our analysis of power reliability.

## Data availability

The climate model data were downloaded from the fifth phase of the Coupled Model Intercomparison Project website [http://cmip-pcmdi.llnl.gov]. The satellite data from CERES were obtained from website [https://ceres.larc.nasa.gov/order_data.php]. Source data are provided with this paper.

## Code availability

Matlab code for calculating the analytical solutions of power reliability sensitivity is available at [https://github.com/jy8/solar]; other codes are available upon request.

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

## Acknowledgements

We would like to thank Professors Robert Socolow and Tiejian Li for their constructive comments on this work. J.Y. acknowledges support from the National Natural Science Foundation of China (41877158, 51739009), NUIST startup funding (1441052001003), Jiangsu distinguished faculty program, and NUIST's supercomputing center. A.P. acknowledges support from the USDA Agricultural Research Service cooperative agreement 58-6408-3-027; and National Science Foundation (NSF) grants EAR-1331846, EAR-1316258, FESD EAR-1338694, and the Carbon Mitigation Initiative at Princeton University. A.M. acknowledges support from the Khalifa University Competitive Internal Research Award, CIRA-2018-102.

## Author contributions

J.Y., A.M., and A.P. conceived and designed the study. J.Y. wrote an initial draft of the paper, to which all authors contributed edits throughout.

## Competing interests

The authors declare no competing interests.
