## [Peer Review File · Nature Communications]

Reviewers' comments:

Reviewer #1 (Remarks to the Author):

The authors put forward the impacts of solar irradiance variations in relation to the future PV reliability. The intermittent character of solar and wind power is well known. It is not clear if this work will aid scientists since even nowadays large storage projects are increasing in both number and capacity, substantially reducing the impact of solar variability. Forecasting solar irradiance and wind speed hours or days ahead aids the system operators for stable grid operation. Thus, I invite the authors to revise their manuscript to address specific concerns detailed below.

This work is practically based upon climate model outputs, but validation data are not detailed. This is clearly important since otherwise this work is not justified. The reader should not search other papers to consider the differences between the models, what were the input parameters, their importance, which is the sensitivity of the models to those parameters, why the time span 2041-2050 was chosen and so on.

For example, are the same conclusions reached if one would examine the 2031-2040 or 2021-2030 time periods?

Line 62. The adopted Kd values are not detailed to the reader.

Line 99. "While the solar radiation in Europe is projected to decrease in January and increase in July, the radiation in Middle East decreases in both months.". Explain why this happens. (also related to Line 195 below)

Line 119. Section "Quantifying sensitivity of power reliability to climate change". The authors state "To systematically assess this linkage, we consider in detail satellite data as well as climate model outputs under the historical climate conditions (see dark color curves in Figure 3a and Supplementary Fig. 3)." It is important for the reader to understand the physical causes behind this behavior of the parameters and not just see plots of some parameters.

Line 133. Fig. 3. Figure captions should not be such detailed like normal text in the body of the manuscript.

Line 166. Fig. 4. Given the nature of the paper and the uncertainty in the climate models, it is not necessary to quote numbers to the third decimal.

Line 195 "Discussion". "... power reliability is less sensitive to mean solar radiation in wet regions (e.g., Europe, Southeast China, Southeast United States) and more sensitive in dry regions (Middle East and Northern Africa)". It is important to explain this performance and not just quoted. Such information is important to see if mitigation measures can be considered.

Line 342. The authors refer to "For a power system with daily storage capacity, LOLP" but no data are given about the battery model since this is directly related to the input energy (from Sun-PV) and output energy (to the loads). Clarify.

Section "Methods". Temperature typically accounts for, roughly, a less than 10% annual energy loss depending on the site. However, since this work performs a global search it should consider temperature effects. Roughly speaking, a 5-7% change between south African countries and northern European countries is expected in normalized energy production.

Minor

Supplementary Fig. 3. The use of the same color does not aid the reader to easily identify the different curves

Line 78. Change "panels" to "modules"

Reviewer #2 (Remarks to the Author):

Review of the paper "Impacts of Solar Intermittency 1 on Future Photovoltaic Reliability" by Yin and co-authors, submitted for publication in Nature Communications.

The paper tackles the problem of solar energy availability in future climate, assessing the relative importance of changes in mean and intermittence for photovoltaic power reliability at global scale. Yin and co-authors highlight that the sensitivity of power reliability to fluctuations of mean solar radiation is geographically inhomogeneous. In particular, regions with the highest solar insolation will be affected by largest fluctuations and reduced reliability.

The paper is well written in a clear and concise manner, figures are effective in delivering key information. The methodology is well described, mathematics and statistics are properly used, and conclusions follow from evidence. The paper has value, especially in the context of the current and future climate change, with renewable energy playing a crucial role in the effort for mitigating global warming. However, before recommending the paper for publication, I believe that some important issues in the study should be addressed.

Major point

The estimation of future loss-of-load-probability sensitivity (L_s), which gives an indication of the photovoltaic power reliability, depends on choice of the pdf shape, in this case the beta function. However, the choice of using beta function at the global scale does not appear fully convincing.

Two papers cited as reference in the Methods section (Bendt et al. 1981; Hollands et al. 1983) actually focus on the US. In Fig. 1, two very specific situations are showed. In Fig. 3 and S3, pdfs of daily clearness index (K) in different regions are displayed, without any indication of the regions where PDFs are derived. This way of presenting PDFs has the merit of being visually clear, but it cannot be used to claim that beta function is the correct fit for K pdfs globally.

The authors should provide quantitative evidence for the pdf choice. I suggest to perform a simple goodness-of-fit test at any location, showing that the beta function is a good approximation at the global scale. To make things simpler, the authors could perform the test at the regional scale, but regions must to be selected carefully to represent the main climatic zones on Earth (e.g. Tropics, subtropics, mid-latitudes). Testing at least another pdf shape would be ideal.

Minor points

Line 17: I find the wording 'nonlinear' misleading in this context. What you show is a geographically heterogeneous response of L_s to changes in mean solar radiation, therefore you should actually emphasize the 'geographical' aspect rather than the 'linearity' of the response. Please make a check across the text for similar wordings.

L. 28: check the currency.

L. 51-53: the motivation for showing Romania and UAE as examples looks quite weak. Please elaborate more, showing how these two examples are interesting, focusing on climate considerations if possible.

L. 65-76: I'd move these lines in the introductory section.

Figure 1: please elaborate a bit on the climate differences between Romania and UAE (evident in the K distribution).

Figure 2: could you also provide the degree of agreement among climate models on the sign of the change? Same for Fig. S1 and S2.

Figure 3: mention the regions used for computations and the motivation for choosing them.

Equation 2: why do you add 100% in the formulation?

L. 162: Please check the references to Figure 4.

Figure 4: how southeast Europe is defined?

L. 177-186: instead of discussing the example of Southern Romania only on the basis of a reference, why not to use climate models to produce an additional figure showing the actual impact on future LOLP at the global scale?

L. 195: referring to cloud climatology pattern comes out of the blue at this point. You only give a reference and no way to compare with Fig. 5, also the relationship with Ls is unclear. Please elaborate more.

L. 196: the wording 'wet regions' is inaccurate. For instance Europe and Southeast China are not wet in the same way. You should refer to climatic zones, such as Tropics, monsoonal regions, subtropics, mid-latitudes.

L. 199-200: your statement about wet regions is too simplistic. For instance, while subtropics are actually projected to be sunnier in the future, higher latitudes are expected to be cloudier because of the shift in the mid-latitude storm track (see e.g. Gaetani et al. 2014, <https://doi.org/10.1016/j.rser.2014.07.041>).

Reviewer #3 (Remarks to the Author):

General comments:

The paper provides valuable insights into the future projections of solar energy production. The authors proposed to use the "Loss-of-load probability" functional to assess the availability of solar energy at future climate conditions. Based on the satellite observations and IPCC model outputs, they were able to make a PV-panel lifetime (30-year) projection that accounts for the intermittency of the renewable energy source. The paper is well written and could be published after a minor revision.

Specific comments:

L50: I think you should talk about the "scattering and absorption" of radiation by all optically active constituents in the atmosphere.

L77-93: Please clarify what data have been used to plot Figure 1. Could you give the estimate of the statistical variability of the mean clearness index?

L99-109: The areas in the dust belt receive the most of solar energy. Do the model account for the change of dust activity associated with climate change?

L110-118: How did you calculate the standard deviation to apply the t-test?

L120-130: The satellite instrument can not measure surface radiative fluxes. The surface fluxes from CERES are based on the column model estimates.

L127: remains > remain

L153: difference > relation

L162: Figure 3d > Figure 4d

Figure 4: The relative change of the mean clearness index in Figure 2 is about 1%. It means that in Figure 4, the effect of different sensitivities is small. Please comment on this.

L323-324: The surface fluxes are not directly measured from the satellite.

L334-336: Dust in the atmosphere will affect incoming solar radiation. This can not be controlled by regular maintenance.

Response to Reviewers

Reviewer #1.....	2
Reviewer #2.....	12
Reviewer #3.....	22
References.....	25

Reviewer comment (italicized) is followed by a response.

Reviewers' comments:

Reviewer #1 (Remarks to the Author):

The authors put forward the impacts of solar irradiance variations in relation to the future PV reliability. The intermittent character of solar and wind power is well known. It is not clear if this work will aid scientists since even nowadays large storage projects are increasing in both number and capacity, substantially reducing the impact of solar variability. Forecasting solar irradiance and wind speed hours or days ahead aids the system operators for stable grid operation. Thus, I invite the authors to revise their manuscript to address specific concerns detailed below.

We thank the reviewer for his/her insightful and constructive comments.

In general, we agree with the reviewer that power storage and weather forecast can substantially reduce the impacts of intermittent solar power.

However, these solutions apply to existing plants under stationary climatic conditions, where the system operator is called to use solutions like storage, curtailment, or load shaping to mitigate short-term (from few seconds to days) effects of weather variability and intermittency (*Ela et al. 2013; Perez et al. 2016*).

In contrast, having simple and robust statistics for the assessment of solar resources at decadal timescales, such as the one proposed here, could help us improve the long-term performance of PV-plants, optimize grid-design and operation, and analyze the tradeoff between mitigation practices (power storage, curtailment, load-shaping, geographic dispersion etc.) and reliability.

In the revised manuscript, we stressed the importance of prediction in managing the impact of variability, at least in the short term. We also emphasized the importance of power storage and explained the unique values of our long-term statistical analysis of solar radiation.

This work is practically based upon climate model outputs, but validation data are not detailed. This is clearly important since otherwise this work is not justified. The reader should not search other papers to consider the differences between the models, what were the input parameters, their importance, which is the sensitivity of the models to those parameters,

Thanks for the comment. We have better justified our work and detailed our validation. First of all, we better explained the **implicit bias corrections** already done in the previous version.

Climate model may exhibit systematic biases which usually need to be corrected before being used for climate change impact assessment (*Stocker 2014*). One of the basic bias-correcting methods is to correct the model outputs by using the differences between historical reference data from the model and observations (*Hawkins et al. 2013*). This bias correction was implicitly conducted in our analysis as we focus on the change of solar radiation (rather than the absolute values) from climate model outputs. This is now better explained and emphasized in the revised manuscript.

Furthermore, in the new version, we provided more information regarding the data accuracy, as the reviewer suggested. To this purpose, we compared the climate model outputs to the satellite-derived products and large-scale reliable datasets. The results show that, while presenting some biases, climate models likely have the same order of accuracy as those reliable data. These results were included in the revised manuscript and reported below.

There are only limited numbers of long-term high-quality observations of solar radiation. For example, Observations of solar radiation from WCRP Baseline Surface Radiation Network (BSRN) collected from 58 stations are highly sparse. Other observational networks are extremely local in nature (e.g., Atmospheric Radiation Measurement Climate Research Facility of US Department of Energy, UO SRML, Sudi Network). Up to date, it is one of the biggest challenges to validate the global solar irradiance and surface energy balance in the climate science community (*Trenberth et al. 2009; Wild et al. 2015*). For this reason, we used the National Solar Radiation Database (NSRDB), which was produced by using ground observations, satellite data, and/or meteorological models. It covers the whole United States and argued to be one of the most reliable datasets for assessing the long-term spatial and temporal variability of the solar resource (*Sengupta et al. 2018*).

Two typical outputs with different assimilation models, METSTAT and SUNY, are achieved in NSRDB (https://rredc.nrel.gov/solar/old_data/nsrdb/) and both are recommended by NREL. We compared SUNY and METSTAT during 2001-2010 when both products are available (see **Figure 1 a** and **b**). Of 1415 sites over the United States (sites with missing data are excluded), the root mean square errors (RMSE) between these two outputs are around 0.05, which may be considered as the systematic biases from NSRDB. When further compared these measurements with satellite data (CERES SYN) in the same locations during the same period (see **Figure 1 c, d, e, and f**), one finds similar ranges of RMSE, suggesting that the satellite products are as accurate as these reliable data.

Figure 1 Comparison of clear index averaged over 2001-2010 in (a, c, and e) January and (b, d, and f) July between (a and b) SUNY and METSTAT outputs from National Solar Radiation Data Base, between (c and d) SUNY and CERES, between (e and f) METSTAT and CERES.

Next, we compared the long-term clearness index from the satellite data and the climate model outputs during 2006-2015 averaged at 280 km equal-area grids over the world (**Table 1**, **Figure 2**, and **Figure 3**). The RMSE for some climate models (e.g., CCSM, GFDL) are similar to these SUNY-METSTAT differences from NSRDB, while for others the RMSE is at least of the same order of magnitude. For this reason, we use climate model outputs to assess the future power reliability in this study but also remind the readers of the uncertainties in the results.

Table 1 Root mean square error (RMSE) between climate model outputs and satellite data as shown in **Figure 2** and **Figure 3**.

	January							July						
	Af.	Asia	Oce.	EU	N.Am.	S.Am.	All	Af.	Asia	Oce.	EU	N.Am.	S.Am.	All
ACCESS	0.07	0.08	0.09	0.04	0.05	0.09	0.07	0.10	0.10	0.03	0.08	0.07	0.07	0.07
BCC	0.04	0.11	0.02	0.10	0.08	0.09	0.07	0.06	0.07	0.10	0.07	0.08	0.08	0.07
CanESM	0.06	0.10	0.04	0.06	0.07	0.10	0.07	0.06	0.07	0.04	0.07	0.08	0.07	0.08
CCSM	0.05	0.09	0.05	0.05	0.06	0.07	0.06	0.06	0.07	0.04	0.07	0.08	0.08	0.06
CMCC	0.07	0.11	0.05	0.13	0.08	0.09	0.08	0.05	0.07	0.04	0.09	0.09	0.06	0.08
CSIRO	0.05	0.11	0.08	0.06	0.06	0.10	0.07	0.05	0.10	0.04	0.06	0.10	0.12	0.08
ECEARTH	0.04	0.10	0.03	0.04	0.06	0.05	0.07	0.08	0.07	0.03	0.04	0.04	0.06	0.07
GFDL	0.05	0.09	0.03	0.04	0.04	0.08	0.06	0.06	0.06	0.02	0.03	0.05	0.10	0.06
INMCM	0.06	0.10	0.05	0.07	0.04	0.08	0.07	0.09	0.09	0.02	0.10	0.06	0.08	0.07
IPSL	0.09	0.11	0.11	0.07	0.07	0.16	0.08	0.13	0.12	0.07	0.14	0.11	0.13	0.09
MPI	0.04	0.08	0.05	0.05	0.06	0.08	0.06	0.06	0.06	0.04	0.05	0.08	0.06	0.07

Figure 2 Comparison of clear index averaged over 2006-2015 in January between satellite data (x-axis) and climate model outputs (y-axis) from different climate model outputs (model names are shown in the upper right of the panel).

Figure 3 As in **Figure 2** but for the month of July.

Why the time span 2041-2050 was chosen and so on. For example, are the same conclusions reached if one would examine the 2031-2040 or 2021-2030 time periods?

Thank you for pointing this out. The period 2041-2050 is chosen to be consistent with the lifespan of solar panels. This was clarified in the revised manuscript.

As shown in **Figure 4** below, the sensitivity of loss-of-load probability (L_s) calculated from the data at various time periods is similar. This is accounted for by the relatively invariant $\mu \sim \sigma$ relationships, as shown in **Figure 5**. The $\mu \sim \sigma$ relationship has long been identified in the solar energy industry and used for sizing the solar system (*Liu and Jordan 1960; Klein and Beckman 1987*). Here, we further tested this relationship under future climates and used it to inform us how the intermittency of solar radiation would change along with the change of the mean radiation (i.e., L_s). We added these figures in the supplementary materials to explain these invariant characteristics.

Figure 4 Sensitivity of loss-of-load probability calculated from climate model outputs as $\Delta LOLP / (\Delta \mu / \mu)$, where Δ refers to the corresponding variable in the future subtracting that in the current condition. The current condition refers to the ‘RCP4.5’ scenario during 2006-2015, and the future condition refers to the ‘RCP4.5’ scenario during (a) 2041-2050, (b) 2021-2030, and (c) 2031-2050.

Figure 5 $\mu \sim \sigma$ relationship in the month of January. The standard deviation, σ , in regions where the monthly mean is around μ (binning interval of 0.05). In top-left panel, the data are from CERES during 2001-2009 (black line) and 2010-2018 (red line); in other panels, the data are from the corresponding climate model outputs from ‘RCP4.5’ scenario during 2006-2015 (black line), 2041-2050 (red dash line), 2021-2030 (blue dot line), and 2031-2040 (green dash-dot line).

Line 62. The adopted K_d values are not detailed to the reader.

We added the detail of K_d . We explain K_d is the value of K that is just enough to satisfy energy demand. The adoption of K_d , along with the distribution of K , allows us to define the loss-of-load probability.

Line 99. “While the solar radiation in Europe is projected to decrease in January and increase in July, the radiation in Middle East decreases in both months.”. Explain why this happens. (also related to Line 195 below)

Thank you for the suggestion. Our research group, in collaboration with Lucarini’s group (University of Reading), has analyzed the changes in rainfall seasonality over the world (Feng *et al.* 2013; Pascale *et al.* 2015, 2016). These results help explain the change in solar radiation. In particular, we quantified rainfall seasonality by relative entropy, which has lower values for more evenly distribution rainfall over the year. Figure 3 of Pascale *et al.* (2016), reported below in **Figure**

6, shows that the relative entropy tends to increase in the Mediterranean regions. The change of seasonality means more rain in winter and less in summer, which explains the decrease in solar radiation in January and increase in July. Note that understanding the climate seasonality is a challenge for climate science because it requires knowledge of the large-scale weather system such as monsoons, the Hadley Cell, mid-latitude/tropic cyclones, and their responses to the rising greenhouse gas concentration.

The decrease in solar radiation in the Middle East may be associated with large-scale circulation (Gaetani et al. 2014), cloudiness trends (Yousef et al. 2020), or the positive Aerosol Optical Depth (AOD) trend already documented over large parts of the Middle East for the period 2001 to 2012 (Klingmüller et al. 2016). We cited these results to explain the change in solar radiation in the revised manuscript.

[REDACTED]

Line 119. Section “Quantifying sensitivity of power reliability to climate change”. The authors state “To systematically assess this linkage, we consider in detail satellite data as well as climate model outputs under the historical climate conditions (see dark color curves in Figure 3a and Supplementary Fig. 3).” It is important for the reader to understand the physical causes behind this behavior of the parameters and not just see plots of some parameters.

Great suggestion. We now provide physical interpretation for the distribution of clearness index as detailed below.

Solar radiation is reflected, scattered, and absorbed by clouds, aerosols, water vapor, and other optically active constituents. Not all of the scattered radiation is lost and part of it eventually arrives at the surface of the earth in the form of diffuse radiation. The diffuse radiation has maximum intensity when the concentrations of optically active constituents are moderate, neither reflecting too much solar radiation back to the outer space or allow most of radiation reaches the Earth surface as direct radiation (see Fig.5 in Liu and Jordan 1960, reported below in **Figure 7**). The diffuse radiation also has the largest variations for moderate concentrations of optically active constituents, whose dynamic behaviors control the scattering of solar radiation. Since global horizontal irradiance (GHI) is the sum of direct and diffuse radiation, it is logical to expect largest variations of GHI or clearness index (i.e., K_t) for moderate monthly mean clearness index (i.e., \bar{K}_t) as presented in Figure 3 of the manuscript.

In the revised manuscript, we cited the reference and explained the figures by using the relationship between direct and diffuse radiations.

[REDACTED]

Line 133. Fig. 3. Figure captions should not be such detailed like normal text in the body of the manuscript.

Done. The caption is shortened and now it reads:

Statistics of clearness index. **(a)** Probability density functions (pdf) of daily clearness index (K) in different regions over the world (binning width of 0.05) from the satellite data in January during 2001-2009 (dark color) and during 2010-2018 (light color). **(b)** Relationship between mean (μ) and standard deviation (σ) of daily K . The black/blue/red dots correspond to the lines in the panel (a); the grey dots are from 11 climate model outputs during 2006-2015; the dash green curve shows the best quadratic fit. **(c)** $d\sigma/d\mu$ calculated from the derivative of the corresponding σ vs μ relationships in panel (b).

Line 166. Fig. 4. Given the nature of the paper and the uncertainty in the climate models, it is not necessary to quote numbers to the third decimal.

Done. Thank you for the reminder.

Line 195 "Discussion". "... power reliability is less sensitive to mean solar radiation in wet regions (e.g., Europe, Southeast China, Southeast United States) and more sensitive in dry regions (Middle East and Northern Africa)". It is important to explain this performance and not just quoted. Such

information is important to see if mitigation measures can be considered.

We agree. We did not explain this point very well in the original manuscript, but now have clarified it in the revised manuscript. To explain the geographical patterns, we analyzed the behaviors of the reliability sensitivity function (see **Figure 4** above). This sensitivity, expressed as a function of μ and LOLP at the design period, has higher absolute value for larger μ . These characteristics suggest higher sensitivity of power reliability in regions with lower cloud optical depth (e.g., Middle East and Northern Africa) and lower sensitivity of power reliability in regions with higher cloud optical depth (e.g., Europe, Southeast China, and Southeast United States).

Line 342. The authors refer to “For a power system with daily storage capacity, LOLP” but no data are given about the battery model since this is directly related to the input energy (from Sun-PV) and output energy (to the loads). Clarify.

We clarified this and referred to a popular PV software, PVGIS (<https://ec.europa.eu/jrc/en/pvgis>), which includes the solar radiation input, the energy conversion efficiency, the state of the battery, and loads (Huld et al. 2017; Huld 2017). One of the output variables “percentage of days the battery became fully discharged” has been used to quantify the reliability of the PV system and is the same metrics “LOLP” used in our study. We have revised the manuscript to reflect this point.

Section “Methods”. Temperature typically accounts for, roughly, a less than 10% annual energy loss depending on the site. However, since this work performs a global search it should consider temperature effects. Roughly speaking, a 5-7% change between south African countries and northern European countries is expected in normalized energy production.

The reviewer is right. Temperature influences the energy conversion efficiency and can have significant impacts on power generation in hot climates (Dubey et al. 2013). It is estimated that photovoltaic power output reduces by 0.45% for each degree increase in temperature (Fell et al. 2015; Patt et al. 2013). Therefore, we may treat the temperature rising as equivalent to the increase of power requirement in our original framework and redefine the parameter K_D as

$$K_D = [1 + \gamma_T(T - T_r)]K_{D,r}, \quad (1)$$

where the temperature factor, γ_T , is about 0.0045, T_r is the reference temperature, and $K_{D,r}$ is the specific value of K that is just enough to generate the demanding energy at the reference temperature. With this change, the corresponding LOLP becomes,

$$\text{LOLP} = \int_0^{[1+\gamma_T(T-T_r)]K_{D,r}} f(K) dK. \quad (2)$$

The change of LOLP from current to future climate conditions can be expressed as

$$\Delta \text{LOLP} \approx \underbrace{\mu \frac{\partial \text{LOLP}}{\partial \mu}}_{L_\mu} \left(\frac{\Delta \mu}{\mu} \times 100\% \right) + T \underbrace{\frac{\partial \text{LOLP}}{\partial T}}_{L_T} \left(\frac{\Delta T}{T} \times 100\% \right), \quad (3)$$

where

$$L_T = TK_{D,r} \gamma_T f(K_D). \quad (4)$$

This expression suggests that the change of LOLP has two parts. This first part has been analyzed in the original manuscript and the second part can also be obtained analytically once we have the distribution of K . The sensitivity for temperature, L_T , is always positive (see Eq. (4)), meaning that rising temperature increases the loss-of-load probability.

It should be noted that the temperature impacts on photovoltaic power generation appear much weaker than the solar radiation impacts over the lifespan of photovoltaic modules (*Gaetani et al. 2014*). However, the rising greenhouse gas concentration may increase the near-air temperature in most of the regions over the world, thus reducing the photovoltaic power generation and the power reliability.

We added these analyses in the Methods section and discussed the impacts of rising temperature on power reliability in the manuscript.

Minor

Supplementary Fig. 3. The use of the same color does not aid the reader to easily identify the different curves

We now have a separate panel for each climate model, as shown here in **Figure 5** (see the figure above) and added it to the supplementary material.

Line 78. Change “panels” to “modules”

Done.

We thank the reviewer again for the constructive comments.

Reviewer #2 (Remarks to the Author):

Review of the paper “Impacts of Solar Intermittency I on Future Photovoltaic Reliability” by Yin and co-authors, submitted for publication in Nature Communications.

The paper tackles the problem of solar energy availability in future climate, assessing the relative importance of changes in mean and intermittence for photovoltaic power reliability at global scale. Yin and co-authors highlight that the sensitivity of power reliability to fluctuations of mean solar radiation is geographically inhomogeneous. In particular, regions with the highest solar insolation will be affected by largest fluctuations and reduced reliability. The paper is well written in a clear and concise manner, figures are effective in delivering key information. The methodology is well described, mathematics and statistics are properly used, and conclusions follow from evidence. The paper has value, especially in the context of the current and future climate change, with renewable energy playing a crucial role in the effort for mitigating global warming. However, before recommending the paper for publication, I believe that some important issues in the study should be addressed.

We thank the reviewer for the positive comments and encouragement. We have used these suggestions to improve the manuscript as addressed below.

Major point

The estimation of future loss-of-load-probability sensitivity (L_s), which gives an indication of the photovoltaic power reliability, depends on choice of the pdf shape, in this case the beta function. However, the choice of using beta function at the global scale does not appear fully convincing.

Two papers cited as reference in the Methods section (Bendt et al. 1981; Hollands et al. 1983) actually focus on the US. In Fig. 1, two very specific situations are showed. In Fig. 3 and S3, pdfs of daily clearness index (K) in different regions are displayed, without any indication of the regions where PDFs are derived. This way of presenting PDFs has the merit of being visually clear, but it cannot be used to claim that beta function is the correct fit for K pdfs globally.

The authors should provide quantitative evidence for the pdf choice. I suggest to perform a simple goodness-of-fit test at any location, showing that the beta function is a good approximation at the global scale. To make things simpler, the authors could perform the test at the regional scale, but regions must to be selected carefully to represent the main climatic zones on Earth (e.g. Tropics, subtropics, mid-latitudes). Testing at least another pdf shape would be ideal.

We followed the reviewer’s suggestion and conducted goodness-of-fit tests for the clearness index over three main climatic zones, tropics, subtropics, and temperate in both January and July as shown in **Figure 8** and **Table 2**. Using the maximum likelihood estimation, we fitted of clearness index data with beta distributions, which shows certain seasonal variations in the locations at the temperate and subtropics. The p values of the Kolmogorov-Smirnov goodness-of-fit tests are larger than the significance level of 0.05, failing to reject the null hypothesis that the clearness index data come from beta distributions.

It is also important to note that this beta distribution is a parsimonious choice which we prefer to other unbounded ones (e.g., Weibull and extreme value distributions) used in the literature (e.g.,

Markvart et al. 2006; Kaplani and Kaplanis 2012). However, our framework is not limited to the use of beta distribution but can easily adopt other distributions if they appear more suitable in some regions. In the revised manuscript, we added these statistical tests of clearness index for the representative locations; we also commented on the general framework in this study and its application for various types of distributions.

Figure 8 Probability density function of clearness index. The bars are empirical distribution from CERES data in the month of (a, c, e) January and (b, d, f) July around (a, b) Paris, France, (c, d) Qinghai, China, and (e, f) Manus, Brazil, corresponding to the three main climatic zones of temperate, subtropics, and tropics. The black curves are the fits of the beta distributions using the maximum likelihood estimation.

Table 2 p-value of Kolmogorov-Smirnov goodness-of-fit tests for clearness index

Location	Climate Zone	January, p-value	July, p-value
Paris, France	Temperate Oceanic	0.33	0.22
Southern Qinghai, China	Subtropics	0.36	0.44
Manus, Brazil	Tropics	0.39	0.09

Minor points

Line 17: I find the wording ‘nonlinear’ misleading in this context. What you show is a geographically heterogeneous response of Ls to changes in mean solar radiation, therefore you should actually emphasize the ‘geographical’ aspect rather than the ‘linearity’ of the response. Please make a check

across the text for similar wordings.

Thank you for the suggestion. We changed the corresponding text.

L. 28: check the currency.

We converted this to US dollars.

L. 51-53: the motivation for showing Romania and UAE as examples looks quite weak. Please elaborate more, showing how these two examples are interesting, focusing on climate considerations if possible.

Thank you for pushing us to improve the presentation of the manuscript. We explained that Romania and UAE are chosen to show two contrasting climate conditions and drastically different potentials for solar energy production. Romania, located in the temperate continental climatic zone, shows strong seasonal variations of clouds, whereas Dubai, UAE has a hot desert climate with weak seasonality (see **Figure 9** left). The contrasting behaviors of cloud seasonality are also evident in the distribution of clearness index as shown in Figure 1 of the manuscript (reported here in **Figure 9** right). In Romania, the distributions tend to be positively skewed in January and negatively skewed in July; in Dubai, the distributions are similar in both January and July.

The choice of Southern Romania also helps us to link the change of clearness index to the change of climate seasonality in future climates. In our previous work, we quantified rainfall seasonality by relative entropy, which has lower values for more evenly distributed rainfall over the year. Figure 3 of *Pascale et al. (2016)*, reported below in **Figure 10**, shows that the relative entropy tends to increase in the Mediterranean region. The change of seasonality may suggest more rain in winter and less in summer, which explains the decrease of solar radiation in January and increase in July as already shown in Fig. 2 of the manuscript.

In the revised manuscript, we added the cloud seasonality plots in the supplementary material and explained the choice of Southern Romania and Dubai is associated with the contrasting climate conditions.

Figure 9 (left) Monthly cloud fraction averaged over 2001-2010 from CRU TS v 4.04 data (<https://crudata.uea.ac.uk/cru/data/hrg/>) (Right) Distribution of K in Southern Romania and Dubai in January and July.

[REDACTED]

L. 65-76: I'd move these lines in the introductory section.

Good idea. Thanks.

Figure 1: please elaborate a bit on the climate differences between Romania and UAE (evident in the K distribution).

We now present the cloud seasonality in both Romania and UAE to identify the climate differences in the supplementary material of the manuscript (reported here in **Figure 9**). We also explain how climate seasonality change may influence the solar power generation in mid-latitude temperate regions (*Pascale et al., 2016*), while several other factors like changes in large-scale circulation

(Gaetani et al. 2014), cloudiness (Yousef et al. 2020), Aerosol Optical Depth (Klingmüller et al. 2016) and dust transport (Prospero and Lamb 2003) may have an influence in solar energy generation hotspots like the Middle East and North Africa (MENA) region.

Figure 2: could you also provide the degree of agreement among climate models on the sign of the change? Same for Fig. S1 and S2.

Done.

Over eleven climate models, eight or more have the same sign of change are marked as dots in Figure 11. The corresponding maps for Fig S1 and S2 are reported in **Figure 12** and **Figure 13**. These maps are similar to these t-test results in the original manuscript, corroborating our analysis of climate change impacts on solar radiation and power reliability. We added these maps as the supplementary materials in the revised manuscript.

Figure 11 Agreement among climate models for (a, b) $\Delta\mu/\mu$ and (c, d) ΔLOLP in (a, c) January and (b, d) July.

Figure 12 Agreement among climate models for Δ LOLP in (a) January and (b) July for designed LOLP of 0.2

Figure 13 Agreement among climate models for Δ LOLP in (a) January and (b) July for designed LOLP of 0.4

Figure 3: mention the regions used for computations and the motivation for choosing them.

We followed this suggestion and clarified that all regions over the world with monthly mean clearness index ranging from 0.3 to 0.7 with interval of 0.05 were selected to plot the probability density function of K and the $\mu \sim \sigma$ relationship. These statistics are relatively invariant in response to changing climate in different periods (see **Figure 14**). Motivated by these features, we then obtained the analytical expression of the sensitivity of the power reliability, which is the key message of the study.

We did not explain this point very well in the original manuscript, but now have clarified how we plot Figure 3 and showed invariant $\mu \sim \sigma$ relationship in the revised manuscript.

Figure 14 $\mu \sim \sigma$ relationship in the month of January. The standard deviation, σ , in regions where the monthly mean is around μ (**binning interval of 0.05**). In top-left panel, the data are from CERES during 2001-2009 (black line) and 2010-2018 (red line); in other panels, the data are from the corresponding climate model outputs from ‘RCP4.5’ scenario during 2006-2015 (black line), 2041-2050 (red dash line), 2021-2030 (blue dot line), and 2031-2040 (green dash-dot line).

Equation 2: why do you add 100% in the formulation?

Since most previous studies reported changes in solar radiation in percentage, we add 100% to be consistent with other reports. We clarified this in the revised manuscript.

L. 162: Please check the references to Figure 4.

Revised. This reference should be Figure 4d.

Figure 4: how southeast Europe is defined?

This is region 7 from Jerez et al., 2015, which includes Bulgaria, Cyprus, Greece, Hungary and Romania. This allows us to compare our results with Jerez et al., 2015 and also validate the analytical results with those from climate model outputs. We have clarified this in the revised manuscript.

L. 177-186: instead of discussing the example of Southern Romania only on the basis of a reference, why not to use climate models to produce an additional figure showing the actual impact on future LOLP at the global scale?

Thank you for the suggestion. Actually we already did this, but we did not explain this approach very well in the original manuscript.

Figure 2 in the manuscript (also see **Figures 11-13** here) show the climate change impacts on LOLP at global scale using the climate model outputs. These geographical patterns were then explained in a theoretical framework in the rest of manuscript with the assumptions of beta distribution of K and invariant $\mu \sim \sigma$ relationships. Specifically, we revisited our results in line 177-186 for Southern Romania from climate model outputs to justify our proposed theoretical framework.

In the revised manuscript, we also changed the second section titles as “theoretical framework of the power reliability under changing climates” to stress the data-to-theory transition and clarify the logical flow of the manuscript.

L. 195: referring to cloud climatology pattern comes out of the blue at this point. You only give a reference and no way to compare with Fig. 5, also the relationship with L_s is unclear. Please elaborate more.

We thank the reviewer for the suggestion. In the revised manuscript we now clarify that L_s in Figure 5 is calculated from our theoretical framework (visualized in Figure 4a in the manuscript and reported below in **Figure 15**), which is expressed as a function of the monthly mean clearness index. This index accounts for the scattered and absorbed radiation by all optically active constituents in the atmosphere such as clouds. Therefore, L_s maps are associated with cloud climatology.

Figure 15 Analytical solutions of L_s (As in Figure 4a in the manuscript).

L. 196: the wording ‘wet regions’ is inaccurate. For instance Europe and Southeast China are not wet in the same way. You should refer to climatic zones, such as Tropics, monsoonal regions, subtropics, mid-latitudes.

Thank you for the suggestions.

We referred Europe as the Mediterranean climatic zone and temperate continental climatic zone, Southeast China as the humid subtropics, and Middle East as the arid hot regions.

L. 199-200: your statement about wet regions is too simplistic. For instance, while subtropics are actually projected to be sunnier in the future, higher latitudes are expected to be cloudier because of the shift in the mid-latitude storm track (see e.g. Gaetani et al. 2014, <https://doi.org/10.1016/j.rser.2014.07.041>).

We thank the reviewer for providing this useful reference. We cited it to explain the climate shift and its impacts on the solar radiation and power reliability.

Reviewer #3 (Remarks to the Author):

General comments:

The paper provides valuable insights into the future projections of solar energy production. The authors proposed to use the "Loss-of-load probability" functional to assess the availability of solar energy at future climate conditions. Based on the satellite observations and IPCC model outputs, they were able to make a PV-panel lifetime (30-year) projection that accounts for the intermittency of the renewable energy source. The paper is well written and could be published after a minor revision.

We thank the reviewer for the positive comments and valuable suggestions.

Specific comments:

L50: I think you should talk about the "scattering and absorption" of radiation by all optically active constituents in the atmosphere.

The reviewer is right. Clearness index accounts for all optically active constituents in the atmosphere. We revised the text accordingly.

L77-93: Please clarify what data have been used to plot Figure 1. Could you give the estimate of the statistical variability of the mean clearness index?

We use the data from the Clouds and the Earth's Radiant Energy System (CERES) SYN1deg.

To estimate the variability of the mean clearness index, we follow the center limit law to estimate the 95% confidence intervals as $\mu \pm 1.96\sigma / \sqrt{n}$, where n is the sample size. The estimated intervals are reported here in **Table 3** and added in the supplementary material.

Table 3 95% confidence intervals of mean clearness index in the Southern Romania and Dubai, UAE.

	Southern Romania		Dubai, UAE	
	January	July	January	July
Current	0.388 ± 0.001	0.549 ± 0.001	0.569 ± 0.001	0.606 ± 0.0003
Future	0.402 ± 0.001	0.580 ± 0.001	0.595 ± 0.001	0.607 ± 0.0003

L99-109: The areas in the dust belt receive the most of solar energy. Do the model account for the change of dust activity associated with climate change?

Aerosol is a key climate component and remains a significant source of uncertainty in climate modeling largely due to the sparse observations (Su et al. 2013; Boucher et al. 2013). We discussed this uncertainty in the last paragraph of the manuscript.

L110-118: How did you calculate the standard deviation to apply the t-test?

We performed the t-test as

$$t_{n-1} = \frac{\bar{x}}{s / \sqrt{n}}, \quad (5)$$

where the bar refers to the sample mean, n is number of climate models (i.e., 11), s refers to the sample standard deviation and is calculated as

$$s = \sqrt{\frac{\sum_{i=1}^n x_i - \bar{x}}{n-1}}. \quad (6)$$

This statistical test shows the degree of agreement among climate models. To prove this, we also present additional maps, showing the locations where eight or more climate models have the same sign of change (see **Figure 16**). These maps, similar to these t-test results in the original manuscript, were added in the supplementary materials of the revised manuscript.

Figure 16 Agreement among climate models for (a, b) $\Delta\mu / \mu$ and (c, d) ΔLOLP in (a, c) January and (b, d) July.

L120-130: The satellite instrument can not measure surface radiative fluxes. The surface fluxes from CERES are based on the column model estimates.

Agree. To clarify this, we commented that CERES fluxes are based on model estimates and referred these fluxes as the satellite products or satellite data.

L127: remains > remain

Corrected. Thank you!

L153: difference > relation

Corrected.

L162: Figure 3d > Figure 4d

Corrected.

Figure 4: The relative change of the mean clearness index in Figure 2 is about 1%. It means that in Figure 4, the effect of different sensitivities is small. Please comment on this.

As noticed by the reviewer, the climate change impacts on sensitivity, L_s , is small so that we can calculate the change of LOLP as the multiplication of the change of clearness index by the relatively constant L_s . This further justifies the use of first order Taylor expansion to approximate Δ LOLP. We commented this in the Methods section of the revised manuscript.

L323-324: The surface fluxes are not directly measured from the satellite.

We clarified that these fluxes are based on the column model estimates when we first cited these data.

L334-336: Dust in the atmosphere will affect incoming solar radiation. This can not be controlled by regular maintenance.

Thank you for pointing this out. We meant to suggest the dust on the surface PV module (i.e., photovoltaic soiling) can be removed by regular maintenance. We clarified this in the revised manuscript.

References

- Boucher, O., and Coauthors, 2013: Clouds and Aerosols. *Climate Change 2013: The Physical Science Basis. Contribution of Working Group I to the Fifth Assessment Report of the Intergovernmental Panel on Climate Change*, T.F. Stocker et al., Eds., Cambridge University Press, 571–658.
- Dubey, S., J. N. Sarvaiya, and B. Seshadri, 2013: Temperature Dependent Photovoltaic (PV) Efficiency and Its Effect on PV Production in the World – A Review. *Energy Procedia*, **33**, 311–321, <https://doi.org/10.1016/j.egypro.2013.05.072>.
- Ela, E., V. Diakov, E. Ibanez, and M. Heaney, 2013: *Impacts of Variability and Uncertainty in Solar Photovoltaic Generation at Multiple Timescales* (No. NREL/TP-5500-58274). National Renewable Energy Lab.(NREL), Golden, CO (United States).
- Fell, A., and Coauthors, 2015: Input Parameters for the Simulation of Silicon Solar Cells in 2014. *IEEE J. Photovolt.*, **5**, 1250–1263, <https://doi.org/10.1109/JPHOTOV.2015.2430016>.
- Feng, X., A. Porporato, and I. Rodriguez-Iturbe, 2013: Changes in rainfall seasonality in the tropics. *Nat. Clim. Change*, **3**, 1–5, <https://doi.org/10.1038/nclimate1907>.
- Gaetani, M., T. Huld, E. Vignati, F. Monforti-Ferrario, A. Dosio, and F. Raes, 2014: The near future availability of photovoltaic energy in Europe and Africa in climate-aerosol modeling

- experiments. *Renew. Sustain. Energy Rev.*, **38**, 706–716, <https://doi.org/10.1016/j.rser.2014.07.041>.
- Hawkins, E., T. M. Osborne, C. K. Ho, and A. J. Challinor, 2013: Calibration and bias correction of climate projections for crop modelling: An idealised case study over Europe. *Agric. For. Meteorol.*, **170**, 19–31, <https://doi.org/10.1016/j.agrformet.2012.04.007>.
- Huld, T., 2017: PVMAPS: Software tools and data for the estimation of solar radiation and photovoltaic module performance over large geographical areas. *Sol. Energy*, **142**, 171–181, <https://doi.org/10.1016/j.solener.2016.12.014>.
- , M. Moner-Girona, and A. Kriston, 2017: Geospatial Analysis of Photovoltaic Mini-Grid System Performance. *Energies*, **10**, 218, <https://doi.org/10.3390/en10020218>.
- Kaplani, E., and S. Kaplanis, 2012: A stochastic simulation model for reliable PV system sizing providing for solar radiation fluctuations. *Appl. Energy*, **97**, 970–981, <https://doi.org/10.1016/j.apenergy.2011.12.016>.
- Klein, S. A., and W. A. Beckman, 1987: Loss-of-load probabilities for stand-alone photovoltaic systems. *Sol. Energy*, **39**, 499–512, [https://doi.org/10.1016/0038-092X\(87\)90057-0](https://doi.org/10.1016/0038-092X(87)90057-0).
- Klingmüller, K., A. Pozzer, S. Metzger, G. L. Stenchikov, and J. Lelieveld, 2016: Aerosol optical depth trend over the Middle East. *Atmospheric Chem. Phys.*, **16**, 5063–5073, <https://doi.org/10.5194/acp-16-5063-2016>.
- Liu, B. Y. H., and R. C. Jordan, 1960: The interrelationship and characteristic distribution of direct, diffuse and total solar radiation. *Sol. Energy*, **4**, 1–19, [https://doi.org/10.1016/0038-092X\(60\)90062-1](https://doi.org/10.1016/0038-092X(60)90062-1).
- Markvart, T., A. Fragaki, and J. N. Ross, 2006: PV system sizing using observed time series of solar radiation. *Sol. Energy*, **80**, 46–50, <https://doi.org/10.1016/j.solener.2005.08.011>.
- Pascale, S., V. Lucarini, X. Feng, A. Porporato, and S. ul Hasson, 2015: Analysis of rainfall seasonality from observations and climate models. *Clim. Dyn.*, **44**, 3281–3301, <https://doi.org/10.1007/s00382-014-2278-2>.
- , ———, ———, ———, and ———, 2016: Projected changes of rainfall seasonality and dry spells in a high greenhouse gas emissions scenario. *Clim. Dyn.*, **46**, 1331–1350, <https://doi.org/10.1007/s00382-015-2648-4>.
- Patt, A., S. Pfenninger, and J. Lilliestam, 2013: Vulnerability of solar energy infrastructure and output to climate change. *Clim. Change*, **121**, 93–102, <https://doi.org/10.1007/s10584-013-0887-0>.
- Perez, R., M. David, T. E. Hoff, M. Jamaly, S. Kivalov, J. Kleissl, P. Lauret, and M. Perez, 2016: Spatial and temporal variability of solar energy. *Found. Trends Renew. Energy*, **1**, 1–44, <https://doi.org/10.1561/27000000006>.
- Prospero, J. M., and P. J. Lamb, 2003: African droughts and dust transport to the Caribbean: Climate change implications. *Science*, **302**, 1024–1027, <https://doi.org/10.1126/science.1089915>
- Sengupta, M., Y. Xie, A. Lopez, A. Habte, G. Maclaurin, and J. Shelby, 2018: The National Solar Radiation Data Base (NSRDB). *Renew. Sustain. Energy Rev.*, **89**, 51–60, <https://doi.org/10.1016/j.rser.2018.03.003>.
- Stocker, T., 2014: *Climate change 2013: the physical science basis: Working Group I contribution to the Fifth assessment report of the Intergovernmental Panel on Climate Change*. Cambridge University Press,.
- Su, W., N. G. Loeb, G. L. Schuster, M. Chin, and F. G. Rose, 2013: Global all-sky shortwave direct radiative forcing of anthropogenic aerosols from combined satellite observations and GOCART simulations. *J. Geophys. Res. Atmospheres*, **118**, 655–669, <https://doi.org/10.1029/2012JD018294>

- Trenberth, K. E., J. T. Fasullo, and J. Kiehl, 2009: Earth's Global Energy Budget. *Bull. Am. Meteorol. Soc.*, **90**, 311–323, <https://doi.org/10.1175/2008BAMS2634.1>.
- Wild, M., and Coauthors, 2015: The energy balance over land and oceans: an assessment based on direct observations and CMIP5 climate models. *Clim. Dyn.*, **44**, 3393–3429, <https://doi.org/10.1007/s00382-014-2430-z>.
- Yousef, L. A., M. Temimi, A. Molini, M. Weston, Y. Wehbe, and A. Al Mandous, 2020: Cloud Cover over the Arabian Peninsula from Global Remote Sensing and Reanalysis Products. *Atmospheric Res.*, **238**, 104866, <https://doi.org/10.1016/j.atmosres.2020.104866>

REVIEWER COMMENTS

Reviewer #1 (Remarks to the Author):

The current version has been improved following the comments of all reviewers.

I understand that the authors are active in the field of environmental engineering. However, the proposed work inevitably involves the field of electrical engineering. Technology advances and we do expect significant improvements in the near future in the field of energy storage, smart grids, electric cars in a large scale. All this affect the proposed work.

It is clear that the Earth's atmosphere is a dynamic system. It was and will always be one. The authors project weather conditions to the future but do not advance accordingly the ability of storage technologies or other approaches to mitigate weather effects upon energy production. As in other cases, the authors may examine scenarios (business as usual or adopt today's technology on mitigating PV power variability which is the current version of the paper, assume a scenario where e.g. 25% of the variability is mitigated and another scenario where 50% is mitigated). It is expected that global maps of LOLP will be smoother.

Section titles seem to be missing (e.g. Introduction, Data, etc)

Reviewer #2 (Remarks to the Author):

Review of the paper "Impacts of Solar Intermittency on Future Photovoltaic Reliability" by Yin and co-authors, submitted for publication in Nature Communications.

I first thanks the authors for taking into account all my comments and providing detailed responses. The authors addressed all the issues I raised, making new analysis and computations also to respond to the other reviewers' requests, and I feel the manuscript substantially improved and is now ready for publication.

Nonetheless, I'd like to highlight a last minor point that could be still addressed to improve the presentation of the results. In my previous review I asked to perform a goodness-of-fit of the beta function, to make sure this is the best shape on a global basis. The authors presented the results of a Kolmogorov-Smirnov test for 3 locations chosen in different climatic zones. When I asked for a test at regional scale, I actually meant to take averages over large regions, which should be selected to be climatically homogeneous. I understand that this is quite a lot of work and is beyond the scope of the paper, but I think that presenting just three locations, even in different climatic zones, is not really representative. The easiest way to present the goodness-of-fit assessment in a convincing and compact way would be to show a global map with the result of the KS test at each grid point, to make sure that the beta function assumption is verified at the global scale.

Reviewer #3 (Remarks to the Author):

The paper deals with a potentially very important issue that could affect the future of the planet. The solar energy is the most significant renewable resource that allows reducing greenhouse emissions. However, a wise decision making on planning and implementation phase is extremely important. The most significant achievement of this study in my view is accounting for solar flux intermittency and in reducing the dimensionality of the evaluation problem to only mean solar radiation. The approximation of $f(K)$, which is fundamental for this purpose, appears to be quite accurate. And this results will stay. Only this would warrant the publication. The prediction of

LOLP, however, is as good, as is a prediction of solar surface flux. Solar flux changes depend both on change in cloudiness and aerosol radiative effect. Both of these factors are poorly predicted. The authors wisely use the multimodel ensemble to evaluate their changes. All models have to account for the cloudiness effect. What about aerosol prediction? Are all chosen models have interactive aerosols and predict the associated solar deeming? I believe clarifying this minor issue would help to better evaluate the results of this interesting study.

Reviewer comment (italicized) is followed by a response.

Reviewer #1 (Remarks to the Author):

The current version has been improved following the comments of all reviewers.

We thank the reviewer for his/her insightful comments. We are also glad that the reviewer appreciated our effort in improving the paper.

I understand that the authors are active in the field of environmental engineering. However, the proposed work inevitably involves the field of electrical engineering. Technology advances and we do expect significant improvements in the near future in the field of energy storage, smart grids, electric cars in a large scale. All this affect the proposed work.

It is clear that the Earth's atmosphere is a dynamic system. It was and will always be one. The authors project weather conditions to the future but do not advance accordingly the ability of storage technologies or other approaches to mitigate weather effects upon energy production. As in other cases, the authors may examine scenarios (business as usual or adopt today's technology on mitigating PV power variability which is the current version of the paper, assume a scenario where e.g. 25% of the variability is mitigated and another scenario where 50% is mitigated). It is expected that global maps of LOLP will be smoother.

As commented by the reviewer, increasing power storage is an essential approach to mitigate the effects of intermittent solar power. We followed his/her suggestions to provide two scenarios with 25% and 50% reduction of the variability of the clearness index. We then investigated its impacts on the LOLP as reported below.

To address this issue, we proceeded as follows. The impacts of increasing power storage may be modeled by reducing the variability of the clearness index as

$$K_b = \mu + b(K - \mu), \quad (1)$$

where the original clearness index K has mean of μ and standard deviation of σ , and the smoothed clearness index is K_b , whose standard deviation becomes

$$\sigma_{K_b} = b\sigma, \quad (2)$$

where the coefficient b controls the reduction of the variability.

Following the reviewer's suggestion, we set b as 0.75 and 0.5 for two scenarios corresponding to the 25% and 50% variability mitigation. We applied Eq. (2) to recalculate the clearness index from 11 climate model outputs during 2041-2050, which were then used to calculate the LOLP numerically. We showed the change of LOLP with no variability mitigation, 25% mitigation, and 50% mitigation in Figure 1. As can be seen, reducing the variability often leads to the decrease of LOLP and thus more reliable power output. These results explain that power storage may mitigate the impacts of intermittent solar power and its uncertainties related to climate change. Note that power storage can only smooth the variability of the power. Regarding the source, when solar power is significantly reduced as in the Middle East, the LOLP also increases even with the variability mitigation (see Figure 1).

We have updated the manuscript to reflect this analysis, adding the variability mitigation scenarios and discussing the effects of power storage.

Figure 1 Ensemble means of the change of LOLP with (a) no variability mitigation, (b) 25% mitigation, and (c) 50% mitigation from 11 climate model outputs between 2006-2015 and 2041-2050 in the month of January. The LOLP during 2006-2015 (i.e., design LOLP) is set as 0.2.

Section titles seem to be missing (e.g. Introduction, Data, etc)

Thank you for pointing this out. We added the section titles.

Reviewer #2 (Remarks to the Author):

Review of the paper “Impacts of Solar Intermittency on Future Photovoltaic Reliability” by Yin and co-authors, submitted for publication in Nature Communications.

I first thank the authors for taking into account all my comments and providing detailed responses. The authors addressed all the issues I raised, making new analysis and computations also to respond to the other reviewers’ requests, and I feel the manuscript substantially improved and is now ready for publication.

We thank the reviewer for taking his/her time to carefully read our manuscript and provide constructive comments. We are also glad that the reviewer appreciated our work.

Nonetheless, I’d like to highlight a last minor point that could be still addressed to improve the presentation of the results. In my previous review I asked to perform a goodness-of-fit of the beta function, to make sure this is the best shape on a global basis. The authors presented the results of a Kolmogorov-Smirnov test for 3 locations chosen in different climatic zones. When I asked for a test at regional scale, I actually meant to take averages over large regions, which should be selected to be climatically homogeneous. I understand that this is quite a lot of work and is beyond the scope of the paper, but I think that presenting just three locations, even in different climatic zones, is not really representative. The easiest way to present the goodness-of-fit assessment in a convincing and compact way would be to show a global map with the result of the KS test at each grid point, to make sure that the beta function assumption is verified at the global scale.

We agree. It is more robust to perform statistical tests of the averages over large regions in different climatic zones over the world. This analysis is now included in the revised manuscript and reported below.

We followed an independent study from Beck et al. (2018) to classify the climatic zones in the 280-km equal-area grids over the world (see Figure 2a). We performed the Kolmogorov-Smirnov tests of the daily clearness index during 2010-2018 in the month of January/July at each grid point averaging over a relatively large area (e.g., within 1000 km radius) and within the same climatic zone. The results show that beta distributions describe well the clearness index in most regions covering 70% of the world (Figure 2 b, c and Table 1). Beta distributions may not be accurate in the Western Sahara in January and in Australia in July. Other distributions tailored for these regions can be directly incorporated into our proposed theoretical framework to analyze the power reliability and will be the subject of future research. Note that the selection of the distributions does not influence the numerical results presented in the first part of the manuscript which directly use the data to calculate LOLP.

We updated the manuscript to add these statistical test results and discuss the adaptation of other distributions for specific locations.

Figure 2 (a) Classification of Climate Zones following Beck et al., (2018) at 280-km equal area grids. Different colors represent different climatic zones (see the legend and Table 1). **(b and c)** The black dots show that the clearness index comes from beta distributions as confirmed by the Kolmogorov-Smirnov tests at 0.05 significant levels. The clearness index is collected at daily timescale from CERES satellite products (see Methods) during 2010-2018 in the month of January (b) or July (c) at each grid point averaging over an area within 1000 km radius and within the same climatic zone as identified in (a).

Table 1 Summary of Kolmogorov-Smirnov test results as in Figure 2.

Climatic Zones	Climate Zone Code	Total number of grids	Number of grids pass K-S tests in January	Number of grids pass K-S tests in July
Tropical, rainforest	Af	87	86	85
Tropical, monsoon	Am	63	49	43
Tropical, savannah	Aw	220	177	168
Arid, desert, hot	BWh	283	100	179
Arid, desert, cold	BWk	96	84	66
Arid, steppe, hot	BSh	104	57	51
Arid, steppe, cold	BSk	122	92	95
Temperate, dry summer, hot summer	Csa	18	12	8
Temperate, dry summer, warm summer	Csb	9	2	5
Temperate, dry summer, cold summer	Csc	0	0	0
Temperate, dry winter, hot summer	Cwa	54	38	35
Temperate, dry winter, warm summer	Cwb	20	13	12
Temperate, dry winter, cold summer	Cwc	0	0	0
Temperate, no dry season, hot summer	Cfa	66	35	58
Temperate, no dry season, warm summer	Cfb	26	22	20
Temperate, no dry season, cold summer	Cfc	0	0	0
Cold, dry summer, hot summer	Dsa	3	0	2
Cold, dry summer, warm summer	Dsb	9	8	1
Cold, dry summer, cold summer	Dsc	7	3	2
Cold, dry summer, very cold winter	Dsd	0	0	0
Cold, dry winter, hot summer	Dwa	17	15	17
Cold, dry winter, warm summer	Dwb	16	3	15
Cold, dry winter, cold summer	Dwc	34	28	33
Cold, dry winter, very cold winter	Dwd	0	0	0
Cold, no dry season, hot summer	Dfa	23	23	21
Cold, no dry season, warm summer	Dfb	99	95	97
Cold, no dry season, cold summer	Dfc	94	78	93
Cold, no dry season, very cold winter	Dfd	0	0	0
Polar, tundra	ET	40	25	37
Polar, frost	EF	0	0	0
All		1510	1045	1143

Reviewer #3 (Remarks to the Author):

The paper deals with a potentially very important issue that could affect the future of the planet. The solar energy is the most significant renewable resource that allows reducing greenhouse emissions. However, a wise decision making on planning and implementation phase is extremely important. The most significant achievement of this study in my view is accounting for solar flux intermittency and in reducing the dimensionality of the evaluation problem to only mean solar radiation. The approximation of $f(K)$, which is fundamental for this purpose, appears to be quite accurate. And these results will stay. Only this would warrant the publication. The prediction of LOLP, however, is as good, as is a prediction of solar surface flux. Solar flux changes depend both on change in cloudiness and aerosol radiative effect. Both of these factors are poorly predicted. The authors wisely use the multimodel ensemble to evaluate their changes. All models have to account for the cloudiness effect. What about aerosol prediction? Are all chosen models have interactive aerosols and predict the associated solar deeming? I believe clarifying this minor issue would help to better evaluate the results of this interesting study.

We thank the reviewer his/her insightful and constructive comments.

As also commented by the reviewer, modelling aerosols and clouds remains the largest sources of uncertainties in climate system (Boucher et al. 2013). The future aerosol and greenhouse gas emissions in climate models are prescribed as different Representative Concentration Pathways (RCPs) (Moss et al. 2010). Our results are based on an intermediate scenario of RCP45, which projects the declining of aerosols during the 21th century because of the emission controls (Rotstayn et al. 2014). While future aerosol emissions are prescribed in RCPs, not all models include the indirect effects related to the aerosol-cloud interaction (see Table 2), which could have certain impacts on the prediction of solar radiation (Chylek et al. 2016). However, this aerosol-cloud interaction seems to have limited impacts on the relationship between the mean and standard deviation of the radiation (see Figure 3), which is key to our analysis of power reliability.

We have updated the manuscript to clarify the future aerosol scenarios and the inclusion of indirect aerosol effects in climate models.

Table 2. Climate Models used in this study and their inclusion of indirect aerosol effects

Acronyms	Model Institutions	Full Indirect Aerosol Effects
ACCESS1.3	Commonwealth Scientific and Industrial Research Organization, Australia	Yes
BCC-CSM1.1(m)	Beijing Climate Center, China	No
CanESM2	Canadian Centre for Climate Modelling and Analysis, Canada	Yes
CCSM4	National Center for Atmospheric Research, USA	No
CMCC-CMS	Euro-Mediterranean Center on Climate Change, Italy	No
CSIRO-Mk3.6.0	Commonwealth Scientific and Industrial Research Organization, Australia	Yes
EC-EARTH	EC-Earth consortium, Europe	No
GFDL-CM3	NOAA Geophysical Fluid Dynamics Laboratory, USA	Yes
INM-CM4	Institute for Numerical Mathematics, Russia	No
IPSL-CM5A	Institute Pierre Simon Laplace, France	No
MPI-ESM-MR	Max Planck Institute for Meteorology (MPI-M), Germany	No

Figure 3. $\mu \sim \sigma$ relationship in the month of January. The standard deviation, σ , in regions where the monthly mean is around μ (binning interval of 0.05). In top-left panel, the data are from CERES during 2001-2009 (black line) and 2010-2018 (red line); in other panels, the data are from the corresponding climate model outputs from ‘RCP4.5’ scenario during 2006-2015 (black line), 2041-2050 (red dash line), 2021-2030 (blue dot line), and 2031-2040 (green dash-dot line).

References

- Beck, H. E., N. E. Zimmermann, T. R. McVicar, N. Vergopolan, A. Berg, and E. F. Wood, 2018: Present and future Köppen-Geiger climate classification maps at 1-km resolution. *Sci. Data*, **5**, 180214, <https://doi.org/10.1038/sdata.2018.214>.
- Boucher, O., and Coauthors, 2013: Clouds and Aerosols. *Climate Change 2013: The Physical Science Basis. Contribution of Working Group I to the Fifth Assessment Report of the Intergovernmental Panel on Climate Change*, T.F. Stocker et al., Eds., Cambridge University Press, 571–658.
- Chylek, P., T. J. Vogelsang, J. D. Klett, N. Hengartner, D. Higdon, G. Lesins, and M. K. Dubey, 2016: Indirect aerosol effect increases CMIP5 models’ projected arctic warming. *J. Clim.*, **29**, 1417–1428.
- Moss, R. H., and Coauthors, 2010: The next generation of scenarios for climate change research and assessment. *Nature*, **463**, 747–756.
- Rotstayn, L. D., and Coauthors, 2014: Declining aerosols in CMIP5 projections: Effects on atmospheric temperature structure and midlatitude jets. *J. Clim.*, **27**, 6960–6977.

REVIEWERS' COMMENTS:

Reviewer #1 (Remarks to the Author):

The manuscript has been improved over the original version and can be accepted for publication.

Reviewer #2 (Remarks to the Author):

I thank the authors for taking into account my suggestions and performing more detailed analysis to improve the robustness of their findings. Interestingly, the beta function seems not be fitting for subtropical climates, this could be an aspect to investigate in future research. My last and final request is to report the number of successful KS tests as percentage to facilitate the reading of Table S2. I also suggest to add lines reporting the results for macro climatic zones, namely tropical, arid, temperate, cold and polar. I'm now happy to recommend the paper for publication.

Reviewer #3 (Remarks to the Author):

I like the paper and recommend to publish it in its current form

Each reviewer comment (italicized) is followed by a response.

REVIEWERS' COMMENTS:

Reviewer #1 (Remarks to the Author):

The manuscript has been improved over the original version and can be accepted for publication.

We thank the reviewer for taking his/her time to carefully review our manuscript.

Reviewer #2 (Remarks to the Author):

I thank the authors for taking into account my suggestions and performing more detailed analysis to improve the robustness of their findings. Interestingly, the beta function seems not be fitting for subtropical climates, this could be an aspect to investigate in future research. My last and final request is to report the number of successful KS tests as percentage to facilitate the reading of Table S2. I also suggest to add lines reporting the results for macro climatic zones, namely tropical, arid, temperate, cold and polar. I'm now happy to recommend the paper for publication.

Thank you for reviewing our manuscript. We have updated the Table S2 to report the percentage and included the macro climatic zones (see the revised table below).

Supplementary Table 1. Summary of Kolmogorov-Smirnov test results

Climatic Zones	Climate Zone Codes	Total number of grids	Grids passes K-S tests in January		Grids passes K-S tests in July	
			number	(%)	number	(%)
Tropical, rainforest	Af	87	86	98.9	85	97.7
Tropical, monsoon	Am	63	49	77.8	43	68.3
Tropical, savannah	Aw	220	177	80.5	168	76.4
Arid, desert, hot	BWh	283	100	35.3	179	63.3
Arid, desert, cold	BWk	96	84	87.5	66	68.8
Arid, steppe, hot	BSh	104	57	54.8	51	49.0
Arid, steppe, cold	BSk	122	92	75.4	95	77.9
Temperate, dry summer, hot summer	Csa	18	12	66.7	8	44.4
Temperate, dry summer, warm summer	Csb	9	2	22.2	5	55.6
Temperate, dry summer, cold summer	Csc	0	0	-	0	-
Temperate, dry winter, hot summer	Cwa	54	38	70.4	35	64.8
Temperate, dry winter, warm summer	Cwb	20	13	65.0	12	60.0
Temperate, dry winter, cold summer	Cwc	0	0	-	0	-
Temperate, no dry season, hot summer	Cfa	66	35	53.0	58	87.9
Temperate, no dry season, warm summer	Cfb	26	22	84.6	20	76.9
Temperate, no dry season, cold summer	Cfc	0	0	-	0	-
Cold, dry summer, hot summer	Dsa	3	0	0.0	2	66.7
Cold, dry summer, warm summer	Dsb	9	8	88.9	1	11.1
Cold, dry summer, cold summer	Dsc	7	3	42.9	2	28.6
Cold, dry summer, very cold winter	Dsd	0	0	-	0	-

Cold, dry winter, hot summer	Dwa	17	15	88.2	17	100.0
Cold, dry winter, warm summer	Dwb	16	3	18.8	15	93.8
Cold, dry winter, cold summer	Dwc	34	28	82.4	33	97.1
Cold, dry winter, very cold winter	Dwd	0	0	-	0	-
Cold, no dry season, hot summer	Dfa	23	23	100.0	21	91.3
Cold, no dry season, warm summer	Dfb	99	95	96.0	97	98.0
Cold, no dry season, cold summer	Dfc	94	78	83.0	93	98.9
Cold, no dry season, very cold winter	Dfd	0	0	-	0	-
Polar, tundra	ET	40	25	62.5	37	92.5
Polar, frost	EF	0	0	-	0	-
North Temperate Zone (35-66.5N)		494	431	87.3	431	87.3
North Subtropics (23.5-35N)		252	92	36.5	212	84.1
North Tropics (0-23.5N)		322	174	54.0	287	89.1
South Tropics (0-23.5S)		285	260	91.2	160	56.1
South Subtropics (23.5-35S)		130	71	54.6	31	23.9
South Temperate Zone (35-66.5S)		27	17	63.0	22	81.5
All		1510	1045	69.2	1143	75.7

Reviewer #3 (Remarks to the Author):

I like the paper and recommend to publish it in its current form

We are glad the reviewer likes our paper. Thank you!